# STATISTICAL ADVANTAGE OF SOFTMAX ATTENTION: INSIGHTS FROM SINGLE-LOCATION REGRESSION

**O. Duranthon & L. Zdeborová**
Statistical physics of computation laboratory
École polytechnique fédérale de Lausanne (EPFL)

**P. Marion**[*]
Inria, École Normale Supérieure,
PSL Research University

**C. Boyer**
Laboratoire de Mathématiques d'Orsay
Université Paris Saclay
and Institut Universitaire de France

**B. Loureiro**
Departement d'Informatique
École Normale Supérieure, PSL & CNRS

## ABSTRACT

Large language models rely on attention mechanisms with a softmax activation. Yet the dominance of softmax over alternatives (e.g., component-wise or linear) remains poorly understood, and many theoretical works have focused on the easier-to-analyze linearized attention. In this work, we address this gap through a principled study of the single-location regression task, where the output depends on a linear transformation of a single input token at a random location. Building on ideas from statistical physics, we develop an analysis of attention-based predictors in the high-dimensional limit, where generalization performance is captured by a small set of order parameters. At the population level, we show that softmax achieves the Bayes risk, whereas linear attention fundamentally falls short. We then examine other activation functions to identify which properties are necessary for optimal performance. Finally, we analyze the finite-sample regime: we provide an asymptotic characterization of the test error and show that, while softmax is no longer Bayes-optimal, it consistently outperforms linear attention. We discuss the connection with optimization by gradient-based algorithms.

## 1 INTRODUCTION

Large language models (LLMs) have recently reshaped natural language processing, enabling applications ranging from conversational agents and code generation to knowledge-intensive reasoning. At the heart of these models lies the Transformer architecture (Vaswani et al., 2017), where the use of softmax in attention layers has proven remarkably effective. Despite its dominance, it has computational drawbacks due to its quadratic complexity in the sequence length, while being theoretically arduous to study due to the softmax normalization that couples tokens in a complex manner.

For these reasons, numerous alternatives have been proposed. Notably, kernelized attention approximates the softmax function with kernel feature maps (Wang et al., 2020; Luo et al., 2021; Choromanski et al., 2021; Qin et al., 2022), achieving linear complexity in sequence length. In parallel, state-space models (SSMs, Peng et al., 2023; Poli et al., 2023; Gu & Dao, 2024) introduce linear recurrent dynamics with gating to tackle long-context information retrieval. On the other hand, while less studied empirically, linear attention, which is the element-wise linearization of softmax around the origin, has been extensively studied in theoretical works (e.g. Ahn et al., 2023; von Oswald et al., 2023; Bai et al., 2023; Mahankali et al.; Zhang et al., 2024; Lu et al., 2025; Zhang et al., 2025).

Despite substantial research on alternative attention mechanisms, softmax attention remains the dominant architecture in large-scale language models. A leading hypothesis for this empirical success is its superior performance on retrieval tasks, a view supported by a growing body of work (Arora et al., 2024; Aksenov et al., 2024; Chou et al., 2024; Hsieh et al., 2024; Shen et al., 2024; Wang et al., 2025). While we defer a detailed discussion to Section 2, we briefly summarize the representative study of Shen et al. (2024). These authors characterize optimal scaling laws for language

---

[*]Part of this work was done while the author was at the Institute of Mathematics, EPFL.

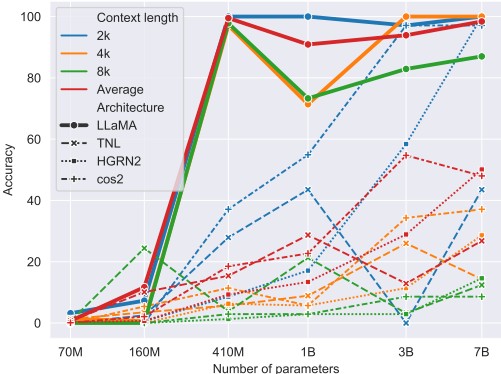 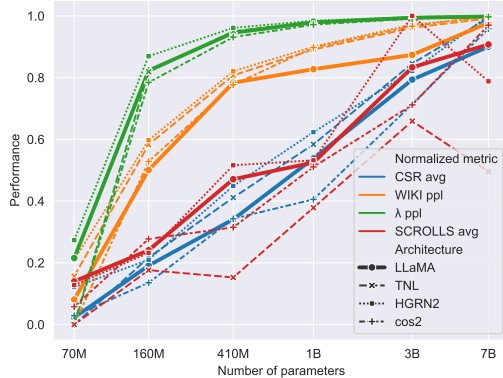

(a) Retrieval tasks (Needle-in-a-Haystack, Kamradt, 2023; Machlab & Battle, 2024). Colors correspond to various context lengths (longer context makes the task harder). The "average" curve is the average over the three tested context lengths.

(b) Linguistic proficiency tasks (accuracy on Common Sense Reasoning and SCROLLS, perplexity on WIKITEXT-2 and LAMBADA datasets). All metrics are normalized so that higher is better, and 1 (resp. 0) corresponds to the best (resp. worse) performance over all tested configurations.

Figure 1 (Shen et al., 2024, Tables 4 and 11): Comparison of softmax Transformer (`LLaMA`, bolded line) with kernelized attention (`TNL`, `cos2`) and state-space models (`HGRN2`), as a function of the model size, and for various tasks (retrieval tasks on the left, linguistic proficiency on the right). All architectures have similar performance for the linguistic proficiency tasks, whereas in retrieval tasks the softmax attention systematically outperforms alternatives.

models equipped with kernelized attention and state-space models (SSMs), and then train softmax-attention, kernelized-attention, and SSM-based models with matched parameter budgets ranging from 70M to 7B. Their downstream evaluation, reproduced for convenience in Figure 1, reveals a striking pattern: although optimally-scaled SSMs and kernelized-attention models achieve competitive performance on linguistic proficiency benchmarks, they consistently underperform softmax attention on retrieval tasks.

Yet the underlying reasons for the advantage of softmax in retrieval tasks remain poorly understood. This gap limits our ability to reason about the tradeoffs and fundamental constraints of different model architectures.

**Contributions and organization.** In this paper, we take a principled approach to the question, with the following contributions. Code to reproduce our simulations is provided.

- We propose a mathematical formalization of tasks where the output depends on a single input token as the single-location regression model (Section 3). This model subsumes previous theoretical studies (Marion et al., 2025), notably by introducing random sequence lengths, and gives a formalization of information retrieval tasks.

- By combining analytical and numerical results, we provide an analysis of the performance of various attention layers, in particular softmax and linear, in increasingly involved settings (approximation, statistical, computational). The analysis remains tractable despite the presence of the softmax nonlinearity by leveraging ideas from the sequence multi-index models (Cui et al., 2024; Arnaboldi et al., 2025; Cui, 2025; Troiani et al., 2025), where properly scaled random variables concentrate in the high-dimensional limit and the learning behavior can then be characterized by a small set of so-called order parameters.

- We prove a gap in approximation performance between linear and softmax attention (Section 4), with the latter reaching Bayes error. We argue that this analysis captures the performance efficiently attainable with one-pass stochastic gradient descent. This gap arises from both the exponential nonlinearity and the normalization in softmax, as illustrated by our comparison with kernelized and element-wise attention (Fig. 2).

- We characterize the regularized empirical risk minimizer (ERM), in the high-dimensional limit with sample complexity linearly proportional to the dimension, as the solution to self-consistent equations (Section 5). By solving these equations, we show that the advantage of softmax over linear attention still holds, and we benchmark against the optimal test risk at fi-

nite sample complexity, which we will call Bayes-optimal performance. We show numerically that the predicted ERM is achieved by gradient-based optimization algorithms (see Fig. 3).

## 2 FURTHER RELATED WORK

**Synthetic information retrieval tasks.** Notable tasks related to single-location include Needle-in-a-Haystack (NIAH) (Kamradt, 2023; Machlab & Battle, 2024), where the goal is to retrieve a fact ("needle") given in a sentence inserted within a large block of unrelated text ("haystack"). An abstraction of NIAH is the Associative Recall (AR) (Graves et al., 2014; Ba et al., 2016) task. In this task, the input contains a sequence of bigrams representing key-value pairs from a random dictionary followed by a query token. For example, the correct output for the input A 2 B 8 C 4 A 3 → B? is 8. This task has been extensively studied through the lens of information retrieval circuits in Transformers, specifically induction heads (Olsson et al., 2022). AR is extended in the Multi-Query Associated Recall (MQAR) task (Arora et al., 2024), featuring several queries for each input sequence.

**Alternatives to softmax attention.** Two main alternative directions feature a linear complexity in the sequence length. First, in kernelized attention (often referred to as linear attention in the literature), the original attention function $\text{Att}(Q, K) = \text{softmax}(QK^\top/\sqrt{d})$ is replaced by $\text{Att}(Q, K) = \varphi(Q)\varphi(K)^\top$, where $\varphi$ is a kernel function (see, among others, Wang et al., 2020; Choromanski et al., 2021; Qin et al., 2022). More recently, state-space models (SSMs) were specifically introduced to handle information retrieval in long contexts (Fu et al., 2023; Peng et al., 2023; Poli et al., 2023; Gu & Dao, 2024). They consist in alternating linear recurrent neural networks (which can be seen as 1d-convolutions along the sequence time) with nonlinear operations applied in parallel over each element of the sequence (sometimes referred to as gating). We refer to Arora et al. (2024) for a common mathematical framework encompassing many variants of SSMs.

While SSMs outperform attention-based models in the specific synthetic task of AR thanks to their capacity to handle long contexts, this success is brittle since they lag behind Transformers on the slightly more involved MQAR task (Arora et al., 2024; Aksenov et al., 2024; Chou et al., 2024; Wang et al., 2025). Arora et al. (2024) propose a theoretical explanation based on the lack of expressivity of convolution-gating models. Transformers also outperform SSMs and kernelized attention in retrieval tasks based on linguistic data (Hsieh et al., 2024; Shen et al., 2024). In this work, we take a principled approach to understanding the benefits of softmax, by identifying an information retrieval framework amenable to theoretical study, and going beyond expressivity results to statistical and computational advantages.

**Methodology for sequence multi-index models.** While our data model is inspired by Marion et al. (2025), it departs significantly by generalizing to variable sequence length, introducing a generic weighting mechanism to encode single location, and renormalizing to obtain proper high-dimensional limits, thereby allowing theoretical analysis of the softmax nonlinearity. The case of diverging signal strength was concurrently studied in Dohmatob (2025). The data model belongs among the sequence multi-index models as introduced in Cui (2025). Special cases of single-index models we studied in Cui et al. (2024); Arnaboldi et al. (2025). Our model corresponds to a sequence two-index case, for which the Bayes-optimal estimator was studied in Troiani et al. (2025). A related classification problem where outputs only depend on a few tokens was concurrently studied in Barnfield et al. (2026) with analysis of steps-wise gradient descent training.

Last, the concurrent work Dragutinović et al. (2025) also discusses advantages of softmax over linear attention, but on a different model (in context classification) and proof techniques.

## 3 TASK AND DATA MODEL

### 3.1 OVERVIEW OF THE SINGLE-LOCATION REGRESSION TASK

We consider a sequence regression task where the input is a sequence $X \in \mathbb{R}^{L \times D}$ of $L$ tokens, each of dimension $D$. The length $L$ of the sequence is allowed to vary, remaining upper bounded by $\bar{L} > 0$. Each sequence is labeled by a scalar $y \in \mathbb{R}$, with the particularity that it depends only on a single input token. We model this single-location dependency by setting a hidden index $\epsilon^* \in \{1, \ldots, L\}$ selecting the relevant token of the input sequence $X$. We emphasize that the latent index $\epsilon^*$ is sample/context-dependent, hence its recovery is akin to a toy in-context learning task.

Without additional structure, there is little hope to retrieve the relevant token. We explore two ways to add such structure, which both involve learning a hidden direction $k^* \in \mathbb{R}^D$ to recover the relevant token. In *spiked single-location regression* (spiked-SLR), a spike is introduced in the direction of $k^*$ at the relevant token $X_{\epsilon^*}$. A closely related scheme, referred to as *max-SLR* (for maximal-correlation SLR), consists in setting $\epsilon^*$ as the index of the token with the largest scalar product with $k^*$.

Our goal is to theoretically study the learning properties of such tasks. For this purpose, we introduce a probabilistic data model that is amenable to analysis in the high-dimensional limit. This framework also unifies the two variants under consideration (spiked-SLR and max-SLR), as described next.

## 3.2  PROBABILISTIC DATA MODEL FOR SLR

Denote by $\mathcal{N}(\omega, V)$ a Gaussian law centered at $\omega$ with covariance $V$, and by $\mathcal{N}(x; \omega, V)$ its density evaluated at $x$. The probabilistic data model begins by drawing two hidden directions

$$k^* \sim \mathcal{N}(0, I_D), \quad v^* \sim \mathcal{N}(0, I_D). \tag{1}$$

Then, each sample $(X, y)$ is drawn as follows. The length $L$ of the sequence is drawn from some discrete law $P_L$ over $\{1, \ldots, \bar{L}\}$, such as uniform or truncated Poisson, while the index $\epsilon^*$ of the relevant token is taken, conditionally on $L$, uniformly at random over $\{1, \ldots, L\}$. The label is then a (potentially random) function of the projection of the relevant token along $v^*$, that is,

$$y = \frac{1}{\sqrt{D}} X_{\epsilon^*} v^* + \Delta \xi, \tag{2}$$

where $\xi$ is independent standard Gaussian noise and $\Delta \geq 0$. It remains to discuss the law of the sequence $X$. A flexible framework consists in taking $X$ as a reweighted Gaussian distribution. More precisely, conditionally on $L, \epsilon^*$ and $k^*$, the density of $X$ at a point $x \in \mathbb{R}^{L \times D}$ is given by

$$P(x | L, \epsilon^*, k^*) = g_\nu(\epsilon^*, \chi^*) \prod_\ell^L \mathcal{N}(x_\ell; 0, I_D), \quad \text{where } \chi^* = \frac{1}{\sqrt{D}} x k^* \in \mathbb{R}^L. \tag{3}$$

Here, $g_\nu : \{1, \ldots, L\} \times \mathbb{R}^L \to \mathbb{R}^+$ is a weight function, indexed by the signal strength $\nu \geq 0$. Its goal is to give more weight to sequences with a large projection $\chi_{\epsilon^*}^* = \frac{1}{\sqrt{D}} x_{\epsilon^*} k^*$. Both the examples from Section 3.1 fall into this framework for specific values of $g_\nu$.

**Spiked-SLR** corresponds to $g_\nu(\epsilon, \chi) = e^{\sqrt{\nu} \chi_\epsilon - \frac{1}{2} \nu}$. Factorizing the density of $x_{\epsilon^*}$ shows that this definition is equivalent to shifting the mean of $X_{\epsilon^*}$ by $\sqrt{\nu} k^*$, while the other tokens are centered, as studied by Marion et al. (2025).

**Maximum-correlation SLR (max-SLR)** is a special case of the sequence multi-index model of Cui (2025); Troiani et al. (2025); Arnaboldi et al. (2025), obtained with $g_\nu(\epsilon, \chi) = L e^{\nu \chi_\epsilon} / \sum_\ell^L e^{\nu \chi_\ell}$. To gain intuition, note that by Bayes' rule, this model is equivalent to first drawing the tokens as independent standard Gaussians, then picking $\epsilon$ randomly with logits proportional to the scalar product of the tokens with $k^*$, that is, $P(\epsilon^* = i | X) = e^{\nu X_i k^*} / \sum_j e^{\nu X_j k^*}$. When $\nu \to \infty$, $\epsilon^*$ corresponds to the index of the token with the largest scalar product with $k^*$.

Other weight functions can be considered as long as they are invariant by label permutation, are properly normalized so that $\int_{\mathbb{R}^D} dx P(x | L, \epsilon^*, k^*) = 1$, and at zero signal $\nu$ are uniform i.e. $g_0(\epsilon, \chi) = 1$.

## 3.3  LEARNING WITH ATTENTION

Since the input sequence may have arbitrary length, we adopt the formalism of working with $(\mathbb{R}^D)_0^{\mathbb{N}}$, the set of sequences in $\mathbb{R}^D$ that are eventually zero. We consider the class of estimators $\mathcal{F}_\sigma = \{f_{\sigma, k, v} : (\mathbb{R}^D)_0^{\mathbb{N}} \to \mathbb{R}\}_{(k, v) \in (\mathbb{R}^D)^2}$, where for a certain activation function $\sigma : \mathbb{R}_0^{\mathbb{N}} \to \mathbb{R}_0^{\mathbb{N}}$ and two vectors $k, v \in \mathbb{R}^D$, the function $f_{\sigma, k, v}$ is defined by

$$f_{\sigma, k, v}(X) = \sigma(\chi)^\top z, \quad \chi = \frac{1}{\sqrt{D}} X k \in \mathbb{R}_0^{\mathbb{N}}, \quad z = \frac{1}{\sqrt{D}} X v \in \mathbb{R}_0^{\mathbb{N}}. \tag{4}$$

We focus on four cases for the activation function $\sigma$. Note that we always assume that $\sigma$ returns $0$ on the vanishing part of the input sequence, so we only define its value on the non-vanishing part.

Table 1: Terminology and notations for the risks. In the text, we use the terms "risk" and "error" in an interchangeable manner.

| | | | |
|---|---|---|---|
| $\mathcal{E}_{\text{Bayes}}$ | Bayes risk (in population) | $\mathcal{E}_{\text{BO}}(\alpha)$ | Bayes-optimal risk (empirical) |
| $\mathcal{E}_\sigma(k, v)$ | population risk of the attention over $(\mathbb{R}^D)^2$ | $\mathcal{L}(k, v)$ | regularized empirical risk (loss) |
| $\tilde{\mathcal{E}}_\sigma$ | population risk parameterized over $\mathbb{R}^7$ or $\mathbb{R}^4$ | $\mathcal{E}_\sigma(\hat{k}, \hat{v})$ | test risk |
| $\mathsf{E}_\sigma$ | minimal population risk | $\mathsf{E}_\sigma(\alpha)$ | minimal test risk |

**Softmax activation** corresponds to the standard choice in current large language models. For an input of length $L$, the softmax writes for $\ell \in \{1, \dots, L\}$ as $\sigma(\chi)_\ell = e^{\chi_\ell} / \sum_{\ell'=1}^{L} e^{\chi_{\ell'}}$.

**Linear activation** writes $\sigma(\chi)_\ell = 1 + \chi_\ell$. We add a constant term to break the symmetry around $\chi = z = 0$, as further discussed in Section 4.2. This corresponds to linearizing the softmax around 0 up to a rescaling. Notice that the activation with the constant term is strictly more expressive than the identity $\sigma(\chi) = \chi$, which can be retrieved by taking large $\chi_\ell$ and small $z_\ell$ at constant $\chi_\ell z_\ell$.

For **element-wise sigmoidal non-linearity**, we focus on $\sigma(\chi)_\ell = 1 + \text{erf}(c + \chi_\ell) = \frac{2}{\sqrt{\pi}} \int_{-\infty}^{c+\chi_\ell} \mathrm{d}x e^{-\frac{1}{2}x^2}$, that varies from 0 to 2, with $c \in \mathbb{R}$ a learnable bias. While for concision we omit this parameter in the mathematical presentation below, it is learned in the numerical experiments. Here also we add a constant term 1 to break the symmetry around $\chi = z = c = 0$.

Finally, we investigate **softplus kernelized attention** $\sigma(\chi)_\ell = \varphi(\chi_\ell) / \sum_{l'} \varphi(\chi_{l'})$ in the particular case of the kernel $\varphi(x) = \text{softplus}(x) = \log(1 + e^x)$.

A limitation of our setting, compared to practical models, is the absence of learnable query vectors, which are not necessary here due to the structure of the task. Still, we note that the output of a standard attention layer for a so-called [CLS] query token exactly corresponds to our model (see Marion et al. (2025) for details). The absence of query vector also means that, in our setting, softmax attention is a special case of kernelized attention by taking $\varphi = \exp$. Our analysis shall still give insightful results on the choice of nonlinearity by comparing softmax to the softplus kernel.

## 4 COMPARISON OF SOFTMAX WITH ALTERNATIVES IN POPULATION RISK

In this section, we compare the *expressivity* of the softmax attention with alternatives on our task, by theoretically assessing the Bayes risk and the minimal population risk for various activation functions. We show that the softmax reaches the Bayes risk. We combine analytical arguments with numerical evidence to support the claim that this approach also captures the performance that is efficiently achievable by running one-pass SGD, which directly minimizes the population risk from random initialization. All proofs of the results in this section are deferred to Appendix A. A summary of the terminology and the notations for the risks is given in Table 1 left.

**Remark 4.1.** *We consider in this section the case $\Delta = 0$ in the output channel (2). Because of independence of the output noise with the other random variables, results would be identical for a positive $\Delta$, up to increasing all errors by $\Delta^2$ corresponding to the irreducible noise.*

### 4.1 BAYES RISK AND OPTIMALITY OF SOFTMAX

In what follows, we assess the performances of the different architectures with respect to the *Bayes risk* (or *Bayes error*) $\mathcal{E}_{\text{Bayes}} = \mathbb{E}[(y - \mathbb{E}(y|X, L, k^*, v^*))^2]$, i.e. the best achievable risk given the task defined in Section 3.2.

**Proposition 4.1.** *Let $L \sim P_L$ and, conditionally on $L$, $\epsilon \sim \text{Unif}(\{1, \dots, L\})$ and $\chi \sim \mathcal{N}(0, I_L)$. Then the Bayes risk is given by*

$$\mathcal{E}_{\text{Bayes}} = 1 - \mathbb{E}_L \mathbb{E}_{\epsilon, \chi} \frac{g_\nu(\epsilon, \chi)^2}{\sum_{\epsilon'=1}^{L} g_\nu(\epsilon', \chi)}. \tag{5}$$

Two particular cases are of interest. First, if the function $g_\nu$ is identically equal to 1, then the position of the relevant token is uniformly distributed over $1, \dots, L$ and independent of the tokens $X$. In this case, the Bayes error equals $1 - \mathbb{E}_\ell[1/L] > 0$. This reflects the irreducible noise stemming from the randomness of the informative token. Conversely, if $g_\nu(\epsilon, \xi) = \mathbb{1}_{\epsilon=1}$, the first token always carries the information. We then return to standard noiseless regression, where the Bayes error is null.

We turn our attention to the best reachable theoretical risk over the class of estimators $\mathcal{F}_\sigma$ when $\sigma$ is the softmax function. Surprisingly, this estimator reaches the Bayes error as shown next.

**Proposition 4.2.** *Assume that, for all $L \geq 1$, $(\epsilon, \epsilon') \in \{1, \dots, L\}^2$, and $\chi \in \mathbb{R}^L$,*

$$\frac{g_\nu(\epsilon, \chi)}{g_\nu(\epsilon', \chi)} = e^{c_\nu(\chi_\epsilon - \chi_{\epsilon'})} , \tag{6}$$

*for some constant $c_\nu \geq 0$. Then, for any $k^*, v^* \in \mathbb{R}^d$,*

$$\min_{f_{k,v} \in \mathcal{F}_{\text{softmax}}} \mathbb{E}_{L,\epsilon^*} \mathbb{E}_{(X,y)}((y - f_{k,v}(X))^2) = \mathcal{E}_{\text{Bayes}}.$$

Note that the requirement on $g_\nu$ defining the distribution of $X$ holds in particular for the spiked-SLR and the max-SLR. In consequence, the softmax architectures $\mathcal{F}_{\text{softmax}}$ reach the Bayes error in both settings, and furthermore the proof reveals that the minimum is reached for $k = c_\nu k^*$ and $v = v^*$, corresponding to recovery of the hidden directions. In statistical physics terminology, the softmax attention is said to satisfy the *Nishimori condition*, as detailed in the proof. This is to be contrasted with the performance of other activation functions, which is characterized next.

## 4.2 EXPRESSION OF THE POPULATION RISK FOR ARBITRARY ACTIVATION FUNCTIONS

We now characterize the theoretical *population risk of the attention* $\mathcal{E}_\sigma(k, v) = \mathbb{E}\big[(y - f_{\sigma,k,v}(X))^2\big]$.

**Proposition 4.3.** *The population risk of the attention $(k, v) \mapsto \mathcal{E}_\sigma(k, v)$ can be reparametrized as a function of the following 7 variables, referred to as order parameters:*

$$m_{kk^*} = \frac{1}{D}k^\top k^* \qquad m_{vv^*} = \frac{1}{D}v^\top v^* \qquad m_{kv^*} = \frac{1}{D}k^\top v^* \qquad m_{vk^*} = \frac{1}{D}v^\top k^* \tag{7}$$

$$q_{kk} = \frac{1}{D}k^\top k \qquad q_{vv} = \frac{1}{D}v^\top v \qquad q_{vk} = \frac{1}{D}k^\top v. \tag{8}$$

The first two order parameters, $m_{kk^*}$ and $m_{vv^*}$, quantify the recovery of the hidden directions, while $q_{kk}$ and $q_{vv}$ are the squared norm of the parameters $k$ and $v$. Finally, the cross-correlation terms $m_{kv^*}$, $m_{vk^*}$, and $q_{vk}$ are nuisance parameters. We let $\tilde{\mathcal{E}}_\sigma : \mathbb{R}^7 \to \mathbb{R}$ be the reparametrized risk depending on the order parameters.

**Optimization dynamics and manifold assumption.** In the following, we consider the case where these three nuisance parameters are null, that is, we restrain our analysis to the manifold

$$\mathcal{M} = \{(k, v) \in (\mathbb{R}^D)^2, m_{kv^*} = m_{vk^*} = q_{vk} = 0\}.$$

The intuition behind this restriction is that, in the SLR, the signals $k^*$ and $v^*$ associated to the keys and the values are drawn independent ; and to obtain good performances, the keys and the values should not mix and should focus on $k^*$ or on $v^*$ separately. This simplification is supported by the following observations pertaining to the landscape of the population risk $\mathcal{E}_\sigma$. In practice, we do not have access to minimizers of the risk, but can instead optimize parameters $k$ and $v$ by one-pass SGD on $\mathcal{E}_\sigma$, which corresponds on average to running gradient descent (GD) on the reparametrized risk $\tilde{\mathcal{E}}_\sigma$. In the high-dimensional limit $D \to \infty$, random Gaussian initialization of the parameters lands on $\mathcal{M}$ because the cross-correlations vanish. Then, the manifold is invariant by GD, in the sense that iterates initialized on the manifold remain on it (see Appendix A.2 for details on this statement and the following). At finite $D$, random initialization lands in a neighborhood of $\mathcal{M}$. In this case, we numerically check that $\mathcal{M}$ is stable in the sense that parameters stay close throughout GD. For linear attention, we also show that the risk has only one (local) minimizer on $\mathcal{M}$, which is also a local minimizer of the risk over the whole parameter space, while for softmax attention we know that a global minimizer is on $\mathcal{M}$. Taken together, these facts support that GD on $\tilde{\mathcal{E}}_\sigma$ and thus one-pass SGD on $\mathcal{E}_\sigma$ converges to a minimizer of the risk on $\mathcal{M}$. A formal proof is delicate, as the landscape features other local minima outside of the manifold, as shown in Appendix A.2.6. This is left for future work.

With a small abuse of notation, we let $\tilde{\mathcal{E}}_\sigma(m_{kk^*}, m_{vv^*}, R_{kk}, R_{vv})$ be the reparametrized risk restricted to $\mathcal{M}$, with $R_{kk}^2 = q_{kk} - m_{kk^*}^2$ and $R_{vv}^2 = q_{vv} - m_{vv^*}^2$ the orthogonal components. It turns out that this object is amenable to insightful theoretical analysis. We first write it explicitly.

**Proposition 4.4.** *Let $L \sim P_L$ and, conditionally on $L$, $\epsilon \sim \mathrm{Unif}(\{1, \ldots, L\})$, $\chi \sim \mathcal{N}(0, I_L)$ and $\xi \sim \mathcal{N}(0, I_L)$. Then, for $m_{kk^*}, m_{vv^*}, R_{kk}, R_{vv} \in \mathbb{R}^4$,*

$$\tilde{\mathcal{E}}_\sigma(m_{kk^*}, m_{vv^*}, R_{kk}, R_{vv}) = \mathbb{E}_L \mathbb{E}_{\epsilon, \chi, \xi} g_\nu(\epsilon, \chi) \left[ 1 - 2m_{vv^*} \sigma(m_{kk^*}\chi + R_{kk}\xi)_\epsilon \right. \tag{9}$$
$$\left. + (m_{vv^*}^2 + R_{vv}^2)\sigma(m_{kk^*}\chi + R_{kk}\xi)^\top \sigma(m_{kk^*}\chi + R_{kk}\xi) \right] .$$

A consequence is the characterization of the activation functions $\sigma$ that allow easy learning, in the sense that the gradients at initialization with respect to the recovery order parameters $m_{kk^*}$ and $m_{vv^*}$ are nonzero. If this quantity vanishes, moving away from the initial condition is harder and requires more iterations and thus samples (Ben Arous et al., 2021). At initialization, $m_{kk^*}$ and $m_{vv^*}$ concentrate around $0$ while $R_{kk}$ and $R_{vv}$ concentrate around $1$. At the first order, the gradient at initialization thus equals $\nabla \tilde{\mathcal{E}}_\sigma(0, 0, 1, 1)$. The next result characterizes when this quantity vanishes. In particular, $\sigma(\chi) = \chi$ and $\sigma(\chi) = \mathrm{erf}(\chi)$ impede learning, which is why we add a constant bias.

**Corollary 4.1.** *If the activation function $\sigma$ has a non-vanishing mean, i.e. if $\mathbb{E}_L \mathbb{E}_{\xi \sim \mathcal{N}(0, I_L)} \sigma(\xi)_\ell \neq 0$ for all $\ell$, then*

$$(\partial_{m_{kk^*}} \tilde{\mathcal{E}}_\sigma, \partial_{m_{vv^*}} \tilde{\mathcal{E}}_\sigma)(0, 0, 1, 1) \neq (0, 0) . \tag{10}$$

*If the activation function $\sigma$ is symmetric, i.e. if $\sigma(-x)_\ell = -\sigma(x)_\ell$ for all $\ell$ and all $x \in \mathbb{R}^L$, then*

$$(\partial_{m_{kk^*}} \tilde{\mathcal{E}}_\sigma, \partial_{m_{vv^*}} \tilde{\mathcal{E}}_\sigma)(0, 0, 1, 1) = (0, 0). \tag{11}$$

We next state that on $\mathcal{M}$ the risk of the linear attention admits a unique minimum, which together with Corollary 4.1 guarantees that GD initialized on $\mathcal{M}$ will converge to it.

**Corollary 4.2.** *The population risk $\mathcal{E}_\sigma$ of the linear attention $\sigma(\chi)_\ell = 1 + \chi_\ell$ admits a unique minimizer over $\mathcal{M}$. It is moreover a (local) minimizer over the whole space $(\mathbb{R})^2$.*

### 4.3 COMPARISON BETWEEN LINEAR AND SOFTMAX ATTENTIONS

Letting $\mathsf{E}_\sigma = \min_{\mathcal{M}} \mathcal{E}_\sigma = \min_{\mathbb{R}^4} \tilde{\mathcal{E}}_\sigma$ be the minimum of the population risk, we now compare the *minimal population risk* $\mathsf{E}_{\mathrm{lin}}$ for linear attention $\sigma(\chi)_\ell = 1 + \chi_\ell$ to the one for softmax $\mathsf{E}_{\mathrm{softmax}} = \mathcal{E}_{\mathrm{Bayes}}$, encompassing asymptotic regimes of strong signals $\nu \to \infty$ or long sequences $L \to \infty$.

**Corollary 4.3.** *Consider the spiked-SLR model with input sequences of deterministic length $L$. The minimal risks attained over the manifold $\mathcal{M}$ satisfy*

$$\mathsf{E}_{\mathrm{lin}} = 1 - \frac{L + \nu(L-1)}{L^2 + \nu(L-1)} \sim \frac{L}{L-1} \frac{1}{\nu} \quad \text{while} \quad \mathsf{E}_{\mathrm{softmax}} = e^{-c_L \nu + o(\nu)} \quad \text{as } \nu \to +\infty \tag{12}$$

*with $c_L > 0$ a constant that depends on $L$. Consider then the max-SLR model at $\nu \to +\infty$ where $g_\nu(\epsilon, \chi) = L \mathbb{1}_{\epsilon = \arg\max_\ell \chi_\ell}$ with deterministic $L$. The softmax attention is well-specified while the linear attention is not: the minimal risks attained over $\mathcal{M}$ satisfy*

$$\mathsf{E}_{\mathrm{lin}} = 1 - \mathcal{O}\left(\frac{\log L}{L}\right) \quad \text{as } L \to \infty \quad \text{while} \qquad \mathsf{E}_{\mathrm{softmax}} = 0 . \tag{13}$$

The first part of the corollary shows that on the spiked-SLR for strong signal $\nu \to +\infty$ the risk vanishes for both attentions. However, the softmax model has a better dependence on $\nu$, thereby establishing its superiority over linear attention in this setting. Turning to the dependency on the sequence length $L$, for the max-SLR, the second part of Corollary 4.3 shows a clear separation in the performance of the linear and the softmax attentions: as the sequence length increases the error of the linear attention converges to 1, which is the error of the trivial null predictor, while the softmax attention reaches perfect prediction for all $L$.

The following corollary illustrates the impact of varying sequence lengths on the linear attention.

**Corollary 4.4.** *Consider the max-SLR model at $\nu \to +\infty$ where $g_\nu(\epsilon, \chi) = L \mathbb{1}_{\epsilon = \arg\max_\ell \chi_\ell}$. Let $f(L) = \mathbb{E}_{\chi \sim \mathcal{N}(0, I_L)} \max_{\ell=1}^L \chi_\ell$. Then, the minimal risk attained over $\mathcal{M}$ by linear attention satisfies*

$$\mathsf{E}_{\mathrm{lin}} = 1 - \frac{1 + (\mathbb{E}_L f(L))^2}{\mathbb{E}_L L} \geq 1 - \frac{1 + (f(\mathbb{E}_L L))^2}{\mathbb{E}_L L} . \tag{14}$$

*Consider then SLR at $\nu = 0$ (i.e. $g_\nu(\epsilon, \chi) = 1$). We obtain that $\mathsf{E}_{\mathrm{lin}} = 1 - 1/\mathbb{E}_L L$ and $\mathsf{E}_{\mathrm{softmax}} = 1 - \mathbb{E}_L(1/L)$ and consequently $\mathsf{E}_{\mathrm{lin}} \geq \mathsf{E}_{\mathrm{softmax}}$, with equality when $\mathrm{Var} L = 0$.*

The first part of the result reveals that the variance in the sequence length $L$ hurts linear attention, since the risk for some distribution of $L$ is always worse than the risk associated to the mean value of this distribution. This illustrates a fundamental limitation of linear attention, arising from its poor normalization properties. In Fig. 2, we indeed observe that the performance gap between softmax and linear attention widens when $L \sim \mathrm{Unif}(1, 2, 3)$ compared to $L = 2$, everything else being fixed. We characterize this phenomenon in the second part of Corollary 4.4, for the case of null signal $\nu = 0$, where the position of the relevant token is independent from the law of the tokens. Here again softmax performs better than linear attention whenever $L$ admits some variance.

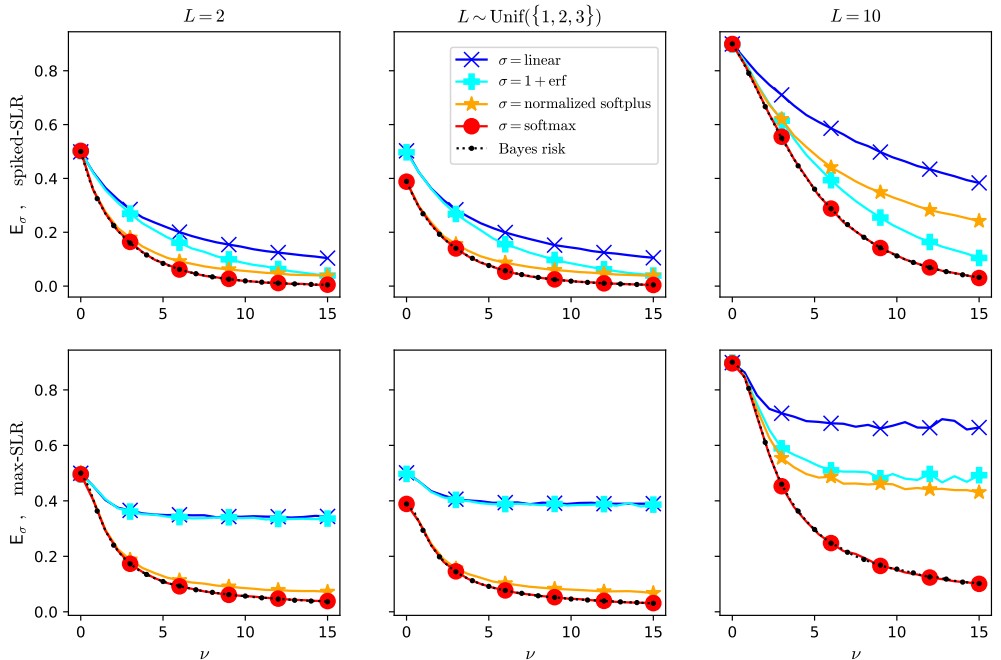

Figure 2: Minimal population risk $\mathsf{E}_\sigma$ over $\mathcal{F}_\sigma$ for different attention activations $\sigma$ (colors), compared to the Bayes risk $\mathcal{E}_{\mathrm{Bayes}}$ (5) (black), for the two tasks spiked-SLR (top) and max-SLR (bottom). Softmax is the only one achieving the Bayes risk. The markers on the lines are for readability only. Population risks are computed via numerical optimization of (9). In all cases, we found that $R_{kk} = R_{vv} = 0$ was optimal, i.e. $k$ exactly aligns with $k^*$ and $v$ with $v^*$.

### 4.4 Other activation functions

We now turn to two non-linear activation functions, element-wise erf and normalized softplus (see Section 3.3). Contrarily to linear attention, the expression (9) cannot be analytically minimized. Instead, we resort to numerical optimization of this expression, and report results in Fig. 2. Overall, we observe that the population risks for these two activation functions are between linear and softmax. In particular, on the max-SLR model the two activation functions are not well-specified. Importantly, the element-wise function suffers from variable sequence lengths while the normalized softplus does not, as can be seen by comparing results for $L = 2$ and $L \sim \mathrm{Unif}\{1, 2, 3\}$. This highlights the importance for the activation function to perform a normalization operation involving all the tokens. Furthermore, the gap between normalized softplus and softmax widens for larger $L$. This can be expected since the softplus does not grow fast enough at $+\infty$ to dominate the noise stemming from the irrelevant tokens. This shows that, for kernelized attention, the kernel has to be well tuned to reach good performances, as is commonly known (see, e.g., Aksenov et al., 2024).

## 5 Performance of the attention at finite sample complexity

So far, our analysis has focused on the properties of the minimizer of the population risk. In practice, however, we only have access to a finite dataset $\mathcal{D} = \left\{ (X_\mu, y_\mu) \in \mathbb{R}^{L(\mu) \times (D+1)} : \mu \in [N] \right\}$, and must therefore rely on an empirical estimator. In this section, we consider the *regularized empirical risk* (or *training loss*) defined for some regularization strengths $r_k \geq 0$ and $r_v \geq 0$ by

$$\mathcal{L}(k, v) = \frac{1}{2} \sum_{\mu=1}^{N} (y_\mu - f_{\sigma,k,v}(X_\mu))^2 + \frac{r_k}{2} \sum_{i=1}^{D} k_i^2 + \frac{r_v}{2} \sum_{i=1}^{D} v_i^2 \,. \tag{15}$$

Our goal is to assess the performance of an empirical risk minimizer (ERM) $(\hat{k}, \hat{v}) \in \arg\min \mathcal{L}$. Our approach here is thus *statistical* in nature: it quantifies the impact of the finite sample size on the performance of attention. We defer computational questions (i.e. how to practically minimize $\mathcal{L}$) to the end of the section. The performance of an estimator $(\hat{k}, \hat{v})$ is evaluated through the *test risk* (or *test error*)

$$\mathcal{E}_\sigma(\hat{k}, \hat{v}) = \mathbb{E}\left[(y - f_{\sigma,\hat{k},\hat{v}}(X))^2 \big| \mathcal{D}, k^*, v^*\right] \tag{16}$$

We investigate the high-dimensional proportional limit where $N, D \to \infty$ at finite $N/D \to \alpha = \Theta(1)$. In this part we take $\alpha$ fixed w.r.t. $N$ and $D$; letting $\alpha \to \infty$ one would recover the population risk discussed in Section 4. For simplicity, we consider the noiseless task $\Delta = 0$ in the following. A summary of the terminology and the notations for the empirical risks is given in Table 1 right.

While in Section 4 the benchmark was the Bayes risk, at finite sample complexity it is natural to consider instead the best achievable test risk conditionally on a finite batch of data $\mathcal{D}$, also known as the *Bayes-optimal risk* (or *Bayes-optimal error*). In the proportional high-dimensional limit, the Bayes-optimal error for the SLR task can be computed by extending the results of Troiani et al. (2025) to random sequence lengths $L \sim P_L$ and arbitrary $g_\nu$. Although this result is of independent interest, for conciseness we refer the reader to Appendix C for the details. The outcome is that the Bayes-optimal error presents a rich phenomenology, with a hard phase where the best-known first-order method fails to achieve the information-theoretical performance, see Section B for a full discussion.

The performance (16) of an ERM is a random quantity that depends on the draw of $k^*, v^*$ and of the dataset. Our main result in this section is that, in the proportional high-dimensional limit and under a concentration assumption known as the replica symmetry, see e.g. Vilucchio et al. (2025), this random variable converges to a deterministic quantity, which can be fully characterized in terms of a few real-valued variables that follow a self-consistent equation. An analogous result for a class of sequence multi-index models was derived in Cui (2025). However, the generality of the result in Cui (2025) did not provide any specific insight about the behavior of the single-location regression model studied here. We instead provide a numerical evaluation of this characterization for our setting.

**Result 5.1** (Test risk for attention-based predictors). *In the proportional high-dimensional limit $N, D \to \infty$ with $\alpha = N/D = \Theta(1)$, under the replica symmetry assumption, the (non-rigorous but standard) replica method predicts that the test risk (16) of a global minimizer $(\hat{k}, \hat{v})$ of the empirical risk (15) converges to a deterministic quantity*

$$\mathcal{E}_\sigma(\hat{k}, \hat{v}) \xrightarrow{\mathbb{P}} \mathsf{E}_\sigma(\alpha) = \min_{(m_k, m_v, q_k, q_v, V_k, V_v) \in \mathcal{S}} \mathbb{E}_L \mathbb{E}_{\xi, \zeta, \chi^*, y} g_\nu(1, \chi^*)(y - \sigma(\gamma)^\top \omega)^2 \tag{17}$$

*where $\gamma = m_k \chi^* + \sqrt{q_k - m_k^2}\xi \in \mathbb{R}^L$, $\omega_1 = m_v y + \sqrt{q_v - m_v^2}\zeta_1$ and, for $\ell > 1$, $\omega_\ell = \sqrt{q_v}\zeta_\ell$. The first expectation is over $L \sim P_L$ and the second conditionally on $L$ over $\xi, \zeta, \chi^* \sim \mathcal{N}(0, I_L)$ and $y \sim \mathcal{N}(0, 1)$ independently. Finally, the set $\mathcal{S} \subset \mathbb{R}^6$ is the set of fixed points of the following iterative self-consistent equations*

$$m_k^{t+1} = (r_k + \hat{V}_k^t)^{-1}\hat{m}_k^t, \qquad q_k^{t+1} = (r_k + \hat{V}_k^t)^{-2}((\hat{m}_k^t)^2 + \hat{q}_k^t), \qquad V_k^{t+1} = (r_k + \hat{V}_k^t)^{-1} \tag{18}$$

$$m_v^{t+1} = (r_v + \hat{V}_v^t)^{-1}\hat{m}_v^t, \qquad q_v^{t+1} = (r_v + \hat{V}_v^t)^{-2}((\hat{m}_v^t)^2 + \hat{q}_v^t), \qquad V_v^{t+1} = (r_v + \hat{V}_v^t)^{-1} \tag{19}$$

$$\begin{pmatrix} \hat{m}_k^t \\ \hat{m}_v^t \end{pmatrix} = \alpha \mathbb{E}_L \mathbb{E}_{\xi, \zeta, y, \chi^*} g_\nu(1, \chi^*) \begin{pmatrix} (V_k^t)^{-1} \sum_\ell^L \left(\chi_\ell^* \chi_\ell' - m_k^t (V_k^t)^{-1} \mathrm{Cov}(\chi_\ell)\right) \\ (V_v^t)^{-1} \left(y z_1' - m_v^t (V_v^t)^{-1} \mathrm{Cov}(z_1)\right) \end{pmatrix} \tag{20}$$

$$\begin{pmatrix} \hat{q}_k^t \\ \hat{q}_v^t \end{pmatrix} = \alpha \mathbb{E}_L \mathbb{E}_{\xi, \zeta, y, \chi^*} g_\nu(1, \chi^*) \sum_{l=1}^L \begin{pmatrix} (V_k^t)^{-2} (\chi_\ell' - \gamma_\ell)^2 \\ (V_v^t)^{-2} (z_\ell' - \omega_\ell)^2 \end{pmatrix} \tag{21}$$

$$\begin{pmatrix} \hat{V}_k^t \\ \hat{V}_v^t \end{pmatrix} = \alpha \mathbb{E}_L L \begin{pmatrix} (V_k^t)^{-1} \\ (V_v^t)^{-1} \end{pmatrix} - \alpha \mathbb{E}_L \mathbb{E}_{\xi, \zeta, y, \chi^*} g_\nu(1, \chi^*) \sum_{l=1}^L \begin{pmatrix} (V_k^t)^{-2} \mathrm{Cov}(\chi_\ell) \\ (V_v^t)^{-2} \mathrm{Cov}(z_\ell) \end{pmatrix} \tag{22}$$

*where we define the potential $\psi_{\mathrm{out}}$ over $(\mathbb{R}^L)^2$, its extremizers and its covariances as:*

$$\psi_{\mathrm{out}}(\chi, z) = -\frac{1}{2}(y - \sigma(\chi)^\top z)^2 + \sum_\ell^L \log \mathcal{N}(\chi_\ell; \gamma_\ell, V_k) + \sum_\ell^L \log \mathcal{N}(z_\ell; \omega_\ell, V_v) \tag{23}$$

$$\chi', z' = \arg\max_{\chi, z} \psi_{\mathrm{out}}(\chi, z) \in (\mathbb{R}^L)^2 \tag{24}$$

$$\text{Cov}\left(\chi_\ell\right) = -\left(\left(\nabla^2 \psi_{\text{out}}(\chi', z')\right)^{-1}\right)_{\chi_\ell} \in \mathbb{R}, \qquad \text{Cov}\left(z_\ell\right) = -\left(\left(\nabla^2 \psi_{\text{out}}(\chi', z')\right)^{-1}\right)_{z_\ell} \in \mathbb{R}. \quad (25)$$

The derivation of this result is given in Appendix C. We simplified it assuming diagonal order parameters, which is analogous to the manifold assumption made in Section 4. Taking the limit $\alpha \to \infty$ we checked that we recover the expression of the population risk of Proposition 4.4, and in particular $\lim_{\alpha \to \infty} \mathsf{E}_\sigma(\alpha)$ corresponds to the value of $\mathsf{E}_\sigma$ derived in Section 4.3 for the population risk. While our derivation is based on the non-rigorous replica method, we expect that a formal proof could be established under the so-called replicon condition along the lines of recent progress in proof techniques from Vilucchio et al. (2025) that is able to deal with minimization of intrinsically non-convex objectives for single-index models. Extending this proof to multi-index models, such as (15), is a technical challenge left for future work.

As in the population result of Proposition 4.3, the high-dimensional analysis shows that the risk depends only on a few order parameters. To give some intuition, they can be interpreted in the following way. $m_k$ and $m_v$ are the alignments of $k$ and $v$ with $k^*$ and $v^*$, respectively. $Q_k$ and $Q_v$ are the squared norms of $k$ and $v$. $V_k$ and $V_v$ are related to the local curvature of the empirical risk $\mathcal{L}$; they encode the fluctuations due to the finite sampling ratio and when $\alpha \to \infty$ we have $V_k, V_v \to 0$. Last, in the self-consistent equations, the potential $\psi_{\text{out}}$ plays the role of an effective low-dimensional empirical risk. Although the resulting expressions are cumbersome, it is important to emphasize that these quantities are deterministic and independent of the diverging dimensions $N$ and $D$, in contrast to the original risk (15). Thus the result provides an implementable formula for the performance of ERMs. In practice, the minimum over the set $\mathcal{S}$ is computed by running the fixed point method from several initializations, typically random (so-called *uninformed*) and *informed* ones corresponding to (partial) alignment of $k$ and $v$ with $k^*$ and $v^*$.

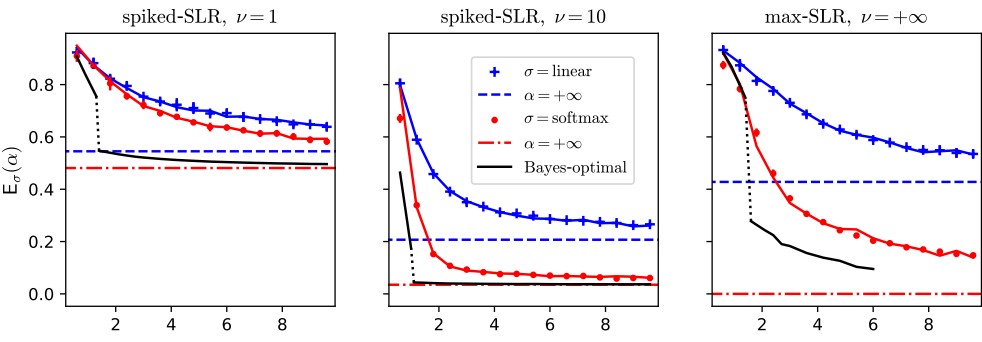

Figure 3: Minimal test risk of the attention (linear vs. softmax) across different tasks and signal strengths $\nu$, for $L = 3$. Linear attention is shown in red and softmax in blue. Solid lines indicate $\mathsf{E}_\sigma(\alpha)$ at finite $\alpha$ (Result 5.1), while markers represent the test risk of an ERM obtained via a local optimization method with $\sqrt{ND} = 10^4$. The regularizations $r_k$ and $r_v$ are tuned by grid search to minimize the test risk, as detailed in Appendix D. Dotted and dashed lines correspond to the value of $\mathsf{E}_\sigma$ in the infinite-$\alpha$ limit (see closed-formed formulas in Proposition 4.1 for softmax and Appendix A.2.7 for linear). The Bayes-optimal risk is shown in black (see Section B for a discussion on its discontinuity). Appendices D-E include more experimental details and results.

We conclude by a few *computational* remarks. Because the empirical risk (15) is non-convex, it is not guaranteed a priori that optimization algorithms can find global minima. To assess the effect of non-convexity, we rely on numerical simulations shown in Fig. 3. More precisely, we compare the prediction of Result 5.1 (here, uninformed and informed initializations give the same result) to the outcome of running a local optimization algorithm (specifically, a quasi-Newton method) on the risk (15), starting from a random initialization, for finite but large $N$ and $D$. We first note that the agreement between both is excellent. This suggests that potential bad local minima in the optimization landscape are avoided, at least for the appropriate regularization strength. Analyzing this landscape is an interesting open question. Furthermore, the softmax attention has lower error than the linear attention across all the tested hyperparameters, which shows that the benefits of softmax extend beyond the population risk analysis of Section 4 to statistical and computational advantages on the empirical risk. We finally note that in this case the softmax is no longer Bayes-optimal but the gap to the Bayes-optimal risk closes as $\alpha$ grows, as expected from our analysis in Section 4.

ACKNOWLEDGMENTS

We would like to thank Hugo Cui for the insightful discussions. This work was supported by the French government, managed by the National Research Agency (ANR), under the France 2030 program with the reference "ANR-23-IACL-0008" and the Choose France - CNRS AI Rising Talents program. P.M. is supported by a Google PhD Fellowship. This work was done in part while P.M. and L.Z. were visiting the Simons Institute for the Theory of Computing. This research was supported by the Swiss National Science Foundation (SNSF) under grant number 212049 (SMArtNet).

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

# A  POPULATION RISK

In this part we prove the expressions for the Bayes risk and the optimal risk of the attention layer, on population loss, stated in Proposition 4.1 and 4.3 and 4.4.

## A.1  PROOF OF PROPOSITION 4.1: BAYES RISK

We consider the Bayes risk $\mathcal{E}_{\text{Bayes}}$. We assume the estimator exactly knows $k^*$ and $v^*$. $\mathcal{E}_{\text{Bayes}}$ is not trivially null because the position $\epsilon^*$ of the relevant token is not known.

The best estimator $\hat{y}_{\text{Bayes}}$ minimizing the square error on a given sample $X \in \mathbb{R}^{L \times D}$ is the posterior mean of $y$ given $L$, $k^*$, $v^*$ and $X$:

$$\hat{y}_{\text{Bayes}} = \sum_\epsilon P(\epsilon|L, k^*, X) y = \sum_\epsilon P(\epsilon|L, k^*, X) \frac{1}{\sqrt{D}} X_\epsilon^\top v^* \tag{26}$$

The conditional distribution of $\epsilon$ given $L$, $k^*$ and $X$ reads

$$P(\epsilon|L, k^*, X) = P(\epsilon|\chi^*) = \frac{g(\epsilon, \chi^*)}{\sum_{\epsilon'=1}^{L} g(\epsilon', \chi^*)} \tag{27}$$

Conditionally on $L$, let $\chi^* = \frac{1}{\sqrt{D}} X k^* \in \mathbb{R}^L$ and $z^* = \frac{1}{\sqrt{D}} X v^* \in \mathbb{R}^L$ be the projections of $X$ onto the two relevant directions. They are distributed according to uncorrelated standard Gaussians with law $\mathcal{N}(0, I_L)$.

We express the empirical means over samples $\mu$ as expectations over one sample. The risk of $\hat{y}_{\text{Bayes}}$ is:

$$\mathcal{E}_{\text{Bayes}} = \mathbb{E}_{L, k^*, v^*} \mathbb{E}_{\epsilon^*, X} (y - \hat{y}_{\text{Bayes}})^2 \tag{28}$$

$$= \mathbb{E}_{L, k^*, v^*} \mathbb{E}_{X \sim \mathcal{N}(0, I_{L \otimes D})} \frac{1}{L} \sum_{\epsilon^*} g_\nu(\epsilon^*, \chi^*)(y - \hat{y}_{\text{Bayes}})^2 \tag{29}$$

$$= 1 + \mathbb{E}_L \mathbb{E}_{\chi^*, z^*} \frac{1}{L} \sum_{\epsilon^*} g_\nu(\epsilon^*, \chi^*) \left[ -2 z_{\epsilon^*}^* \sum_\epsilon P(\epsilon|\chi^*) z_\epsilon^* + \left( \sum_\epsilon P(\epsilon|\chi^*) z_\epsilon^* \right)^2 \right] \tag{30}$$

$$= 1 + \mathbb{E}_L \mathbb{E}_{\chi^*} \frac{1}{L} \sum_{\epsilon^*} g_\nu(\epsilon^*, \chi^*) \left[ -2 P(\epsilon^*|\chi^*) + \sum_\epsilon P(\epsilon|\chi^*)^2 \right] \tag{31}$$

$$= 1 - \mathbb{E}_L \mathbb{E}_{\chi^*} \frac{1}{L} \sum_{\epsilon^*} \frac{g_\nu(\epsilon^*, \chi)^2}{\sum_\epsilon g_\nu(\epsilon, \chi)} \tag{32}$$

This gives the expression stated in result 4.1.

## A.2  RISK OF THE ATTENTION LAYER ON POPULATION LOSS

We consider the risk of the trained attention. We work on population loss. We recall that $k \in \mathbb{R}^D$ and $v \in \mathbb{R}^D$ are the weights of the attention. They can be described by the following scalar order parameters (or summary statistics, or sufficient statistics):

$$m_{kk^*} = \frac{1}{D} k^\top k^* \qquad m_{vv^*} = \frac{1}{D} v^\top v^* \qquad m_{kv^*} = \frac{1}{D} k^\top v^* \qquad m_{vk^*} = \frac{1}{D} v^\top k^* \tag{33}$$

$$q_{kk} = \frac{1}{D} k^\top k \qquad q_{vv} = \frac{1}{D} v^\top v \qquad q_{vk} = \frac{1}{D} k^\top v \tag{34}$$

Setting

$$\left( \begin{smallmatrix} q_{kk} & q_{vk} \\ q_{vk} & q_{vv} \end{smallmatrix} \right) = \left( \begin{smallmatrix} m_{kk^*} \\ m_{vk^*} \end{smallmatrix} \right)^{\otimes 2} + \left( \begin{smallmatrix} m_{kv^*} \\ m_{vv^*} \end{smallmatrix} \right)^{\otimes 2} + \hat{q} \tag{35}$$

with $\hat{q}$ a positive $2 \times 2$ matrix, $k$ and $v$ can be expressed as

$$k = m_{kk^*} k^* + m_{kv^*} v^* + (\tilde{q}^{1/2})_{kk} k^\perp + (\tilde{q}^{1/2})_{kv} v^\perp \tag{36}$$

$$v = m_{vk^*}k^* + m_{vv^*}v^* + (\tilde{q}^{1/2})_{kv}k^\perp + (\tilde{q}^{1/2})_{vv}v^\perp \tag{37}$$

with $k^\perp \in \mathbb{R}^D$ and $v^\perp \in \mathbb{R}^D$ two vectors orthogonal to $k^*$, $v^*$ and between themselves. We introduce the shorthands

$$R_{kk} = (\tilde{q}^{1/2})_{kk} \qquad R_{kv} = (\tilde{q}^{1/2})_{kv} \qquad R_{vv} = (\tilde{q}^{1/2})_{vv} \tag{38}$$

$R_{kk}$, $R_{kv}$ and $R_{vv}$ are related to the magnitude of the components in $k$ and $v$ that are orthogonal to $k^*$ and $v^*$ and bring no information. In the case where there is no overlap between $v$ and $k$ or $k^*$, and no overlap between $k$ and $v$ or $v^*$, i.e. when $m_{kv^*} = m_{vk^*} = q_{vk} = 0$, one can simply express the $R$s as $R_{kk}^2 = q_{kk} - m_{kk^*}^2$, $R_{vv}^2 = q_{vv} - m_{vv^*}^2$ and $R_{kv} = 0$. Then $R_{kk} = R_{vv} = 0$ means that $k$ is perfectly aligned with $k^*$ and $v$ with $v^*$.

The loss depends on $k$ and $v$ only via their projections onto the tokens $X$. Conditionally on $L$, we introduce the projections

$$\chi^* = \frac{1}{\sqrt{D}}Xk^* \qquad\qquad z^* = \frac{1}{\sqrt{D}}Xv^* \tag{39}$$

$$\xi = \frac{1}{\sqrt{D}}Xk^\perp \qquad\qquad \zeta = \frac{1}{\sqrt{D}}Xv^\perp \tag{40}$$

By central limit theorem we have $\chi^* \sim \mathcal{N}(0, I_L), z^* \sim \mathcal{N}(0, I_L), \xi \sim \mathcal{N}(0, I_L), \zeta \sim \mathcal{N}(0, I_L)$. For conciseness we introduce the projections of $k$ and $v$:

$$b = \frac{1}{\sqrt{D}}Xk = m_{kk^*}\chi^* + m_{kv^*}z^* + R_{kk}\xi + R_{kv}\zeta \in \mathbb{R}^L \tag{41}$$

$$a = \frac{1}{\sqrt{D}}Xv = m_{vk^*}\chi^* + m_{vv^*}z^* + R_{kv}\xi + R_{vv}\zeta \in \mathbb{R}^L \tag{42}$$

We introduce

$$\Theta = (m_{kk^*}, m_{kv^*}, m_{vk^*}, m_{vv^*}, R_{kk}, R_{kv}, R_{vv}) \in \mathbb{R}^7$$

the set of parameters over which the loss is effectively minimized. Then the risk reads

$$\mathcal{E}_\sigma(k, v) = \tilde{\mathcal{E}}_\sigma(\Theta) \tag{43}$$

$$\tilde{\mathcal{E}}_\sigma(\Theta) = \mathbb{E}_{L,k^*,v^*}\mathbb{E}_{\epsilon^*,X}(y - \hat{y})^2 \tag{44}$$

$$= \mathbb{E}_{L,k^*,v^*}\mathbb{E}_{X\sim\mathcal{N}(0,I_{L\otimes D}),\epsilon^*}\, g_\nu(\epsilon^*,\chi^*)(y - \hat{y})^2 \tag{45}$$

$$= \mathbb{E}_L\mathbb{E}_{\epsilon^*,\chi^*,z^*,\xi,\zeta}\, g_\nu(\epsilon^*,\chi^*)\left(z^*_{\epsilon^*} - a^\top\sigma(b)\right)^2 \tag{46}$$

and the minimal risk is

$$\mathsf{E}_\sigma = \min_{(k,v)\in(\mathbb{R}^D)^2} \mathcal{E}_\sigma(k,v) \tag{47}$$

$$= \min_{\Theta\in\mathbb{R}^7} \tilde{\mathcal{E}}_\sigma(\Theta) \tag{48}$$

### A.2.1 PROOF OF PROPOSITION 4.2: THE SOFTMAX IS BAYES-OPTIMAL

We recall that the Bayes risk and the optimal risk of the attention estimators are

$$\mathcal{E}_{\text{Bayes}} = 1 - \mathbb{E}_L\mathbb{E}_{\chi^*}\frac{1}{L}\sum_{\epsilon^*}\frac{g_\nu(\epsilon^*,\chi)^2}{\sum_\epsilon g_\nu(\epsilon,\chi)} \tag{49}$$

$$\mathsf{E}_\sigma \leq \min_{m_{kk^*},m_{vv^*}\in\mathbb{R}^2} \mathbb{E}_L\mathbb{E}_{\epsilon,\chi}\left[1 - 2m_{vv^*}g_\nu(\epsilon,\chi)\sigma(m_{kk^*}\chi)_\epsilon + m_{vv^*}^2 g_\nu(\epsilon,\chi)\sigma(m_{kk^*}\chi)^\top\sigma(m_{kk^*}\chi)\right] \tag{50}$$

where for $\mathsf{E}_\sigma$ we have a upper-bound obtained by restraining the min of eq. (48) to $m_{vk^*}, m_{kv^*}, R_{vv}, R_{kk}, R_{kv} = 0, 0, 0, 0, 0$. By definition of the Bayes risk we have $\mathcal{E}_{\text{Bayes}} \leq \mathsf{E}_\sigma$.

We search for the optimal $\sigma$. We show that it has to match the Bayes-optimal estimator of $\epsilon$. We recall that, from the previous part A.1, the posterior distribution of $\epsilon$ given $\chi = \frac{1}{\sqrt{D}}Xk^*$ is

$$P(\epsilon|\chi) = \frac{g(\epsilon,\chi)}{\sum_{\epsilon'=1}^L g(\epsilon',\chi)}. \tag{51}$$

If, for a certain $m_{kk^*}$, $\sigma(m_{kk^*}\chi)_\epsilon = P(\epsilon|\chi)$, then

$$\mathbb{E}_L \mathbb{E}_{\epsilon,\chi}\, g_\nu(\epsilon,\chi)\sigma(m_{kk^*}\chi)^\top \sigma(m_{kk^*}\chi) = \mathbb{E}_L \mathbb{E}_\chi \frac{1}{L}\sum_\epsilon^L g_\nu(\epsilon,\chi)\sum_{\epsilon'}^L P(\epsilon'|\chi)^2 \tag{52}$$

$$= \mathbb{E}_L \mathbb{E}_\chi \frac{1}{L}\sum_\epsilon^L \frac{g(\epsilon,\chi)^2}{\sum_{\epsilon'}^L g(\epsilon',\chi)} \tag{53}$$

$$= \mathbb{E}_L \mathbb{E}_{\epsilon,\chi}\, g_\nu(\epsilon,\chi)\sigma(m_{kk^*}\chi)_\epsilon \tag{54}$$

One can notice the similarity with the derivation of the Bayes risk in part A.1. We proved the equality

$$\mathbb{E}_L \mathbb{E}_{\epsilon,\chi}\, g_\nu(\epsilon,\chi)\sigma(m_{kk^*}\chi)_\epsilon = \mathbb{E}_L \mathbb{E}_{\epsilon,\chi}\, g_\nu(\epsilon,\chi)\sigma(m_{kk^*}\chi)^\top \sigma(m_{kk^*}\chi)\,. \tag{55}$$

In statistical physics this equality is called the Nishimori condition. It is satisfied when $\sigma(m_{kk^*}\chi)_\epsilon$ matches the posterior distribution of $\epsilon$. The optimal $m_{vv^*}$ is then $m_{vv^*} = 1$ and we obtain

$$\mathsf{E}_\sigma = 1 - \mathbb{E}_L \mathbb{E}_{\epsilon,\chi}\, g_\nu(\epsilon,\chi)\sigma(m_{kk^*}\chi)_\epsilon \tag{56}$$

$$= 1 - \mathbb{E}_L \mathbb{E}_\chi \frac{1}{L}\sum_\epsilon^L \frac{g(\epsilon,\chi)^2}{\sum_{\epsilon'}^L g(\epsilon',\chi)} \tag{57}$$

Consequently $\mathsf{E}_\sigma = \mathcal{E}_{\text{Bayes}}$.

It remains to show which $\sigma$ can satisfy the Nishimori condition. We assume that there is a constant $c_\nu$ such that for all $L$, all $\epsilon, \epsilon' \in \{1, \ldots, L\}$ and all $\chi \in \mathbb{R}^L$

$$\frac{g_\nu(\epsilon,\chi)}{g_\nu(\epsilon',\chi)} = e^{c_\nu(\chi_\epsilon - \chi_{\epsilon'})}\,. \tag{58}$$

Then

$$\sigma(m_{kk^*}\chi)_\epsilon = P(\epsilon|\chi) \tag{59}$$

$$= \frac{g(\epsilon,\chi)}{\sum_{\epsilon'}^L g(\epsilon',\chi)} \tag{60}$$

$$= \frac{e^{c_\nu\chi_\epsilon}}{\sum_{\epsilon'}^L e^{c_\nu\chi_{\epsilon'}}} \tag{61}$$

that is to say $\sigma$ is a softmax with inverse temperature $m_{kk^*} = c_\nu$.

### A.2.2 INVARIANT MANIFOLD

In the main part sec. 4 we introduced the manifold

$$\mathcal{M} = \{(k,v) \in (\mathbb{R}^D)^2, m_{kv^*} = m_{vk^*} = q_{vk} = 0\}.$$

We show that it is invariant by GD. We consider the space of the order parameters, parameterized by

$$\Theta = (m_{vv^*}, m_{vk^*}, m_{kv^*}, m_{kk^*}, R_{vv}, R_{kk}, R_{kv}) \in \mathbb{R}^7.$$

We introduce the manifold

$$\tilde{\mathcal{M}} = \{\Theta \in \mathbb{R}^7, m_{kv^*} = m_{vk^*} = R_{kv} = 0\}. \tag{62}$$

At random initialization we have that $\Theta = (0,0,0,0,1,0,1) \in \mathcal{M}$. We consider the dynamics given by the gradient flow over the loss $\tilde{\mathcal{E}}_\sigma$ defined in eq. (44):

$$\dot{\Theta} = -\nabla_\Theta \tilde{\mathcal{E}}_\sigma(\Theta) \tag{63}$$

We show that $\tilde{\mathcal{M}}$ is invariant under this dynamics. The gradient in the three directions $(m_{kv^*}, m_{vk^*}, R_{kv})$ on $\tilde{\mathcal{M}}$ reads

$$\partial_{m_{kv^*}}\tilde{\mathcal{E}}_\sigma = 2\mathbb{E}_L \mathbb{E}_{\epsilon^*,\chi^*,z^*,\xi,\zeta}\, g_\nu(\epsilon^*,\chi^*)\left(a^\top\sigma(b) - z_{\epsilon^*}^*\right) a^\top\nabla\sigma(b)z^* \tag{64}$$

$$\partial_{m_{vk^*}}\tilde{\mathcal{E}}_\sigma = 2\mathbb{E}_L \mathbb{E}_{\epsilon^*,\chi^*,z^*,\xi,\zeta}\, g_\nu(\epsilon^*,\chi^*)\left(a^\top\sigma(b) - z_{\epsilon^*}^*\right) \sigma(b)^\top\chi^* \tag{65}$$

$$\partial_{R_{kv}}\tilde{\mathcal{E}}_\sigma = 2\mathbb{E}_L \mathbb{E}_{\epsilon^*,\chi^*,z^*,\xi,\zeta}\, g_\nu(\epsilon^*,\chi^*)\left(a^\top\sigma(b) - z_{\epsilon^*}^*\right)\left(\sigma(b)^\top\xi + a^\top\nabla\sigma(b)\zeta\right) \tag{66}$$

At $m_{kv^*} = m_{vk^*} = R_{kv} = 0$ we have $a = m_{vv^*}z^* + R_{vv}\zeta$ and $b = m_{kk^*}\chi^* + R_{kk}\xi$ is independent of $z^*$ and $\zeta$. Then, because of the parity with respect to $z^*$ and $\zeta$, we obtain that the gradient vanishes.

A.2.3 STABILITY OF THE MANIFOLD

We show numerically that the manifold

$$\tilde{\mathcal{M}} = \{\Theta \in \mathbb{R}^7, m_{kv^*} = m_{vk^*} = R_{kv} = 0\}$$

is stable. We simulate the gradient flow

$$\dot{\Theta} = -\nabla_\Theta \tilde{\mathcal{E}}_\sigma(\Theta) \tag{67}$$

starting from a perturbed random initial condition $(m_{kk^*}, m_{kv^*}, m_{vk^*}, m_{vv^*}, R_{kk}, R_{kv}, R_{vv}) = (0, 0, 0, 0, 1, 0, 1) + \eta$, with $\eta \sim \text{Unif}([-\bar{\eta}, +\bar{\eta}]^7)$ some small noise. We consider the value of $\Theta$ reached after convergence, for several independent realizations of $\eta$, for all the configurations considered in the main part of the article. On Figure 4 we observe that all the trajectories converge to a point that belongs to $\tilde{\mathcal{M}}$, up to the numerical errors due to the integration. We additionally observe that $R_{kk} \approx R_{vv} \approx 0$.

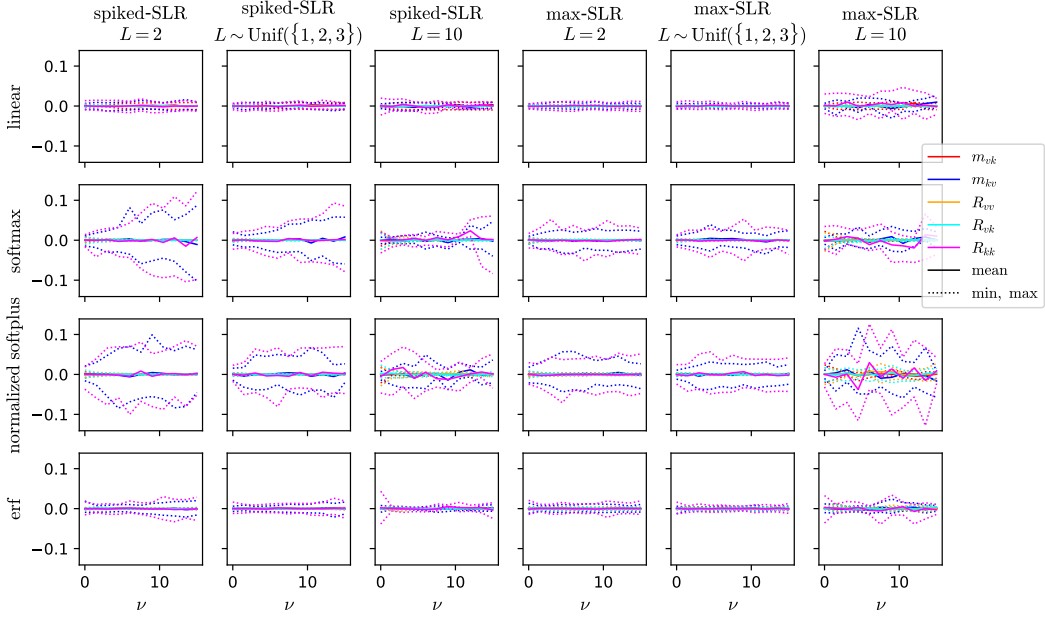

Figure 4: Values of the order parameters $m_{kv^*}, m_{vk^*}, m_{vv^*}, R_{kk}, R_{kv}$ and $R_{vv}$ obtained after convergence of the gradient flow. The mean, max and min are taken over the independent runs. The initial noise is $\bar{\eta} = 0.1$ and there are at least twenty independent runs.

A.2.4 ORTHOGONAL COMPONENTS ON THE MANIFOLD

We show that the minimization of $\tilde{\mathcal{E}}_\sigma$, on the previously defined manifold, leads to a vanishing orthogonal component $R_{vv} = 0$. We can show that the second orthogonal component $R_{kk}$ is null if we assume that the attention is linear.

We recall that on the manifold $a = m_{vv^*} z^* + R_{vv} \zeta$ and $b = m_{kk^*} \chi^* + R_{kk} \xi$. We compute the gradient of the loss with respect to $R_{vv}$:

$$\partial_{R_{vv}} \tilde{\mathcal{E}}_\sigma = 2 \mathbb{E}_L \mathbb{E}_{\epsilon^*, \chi^*, z^*, \xi, \zeta} \, g_\nu(\epsilon^*, \chi^*) \left( a^\top \sigma(b) - z_{\epsilon^*}^* \right) \sigma(b)^\top \zeta \tag{68}$$

$$= 2 R_{vv} \underbrace{\mathbb{E}_L \mathbb{E}_{\epsilon^*, \chi^*, \xi} \, g_\nu(\epsilon^*, \chi^*) \sigma(b)^\top \sigma(b)}_{>0} \tag{69}$$

where we used the parity with respect to $\zeta$ and assumed that the supports of $\sigma$ and $b$ overlap. As a consequence $R_{vv} = 0$ is the minimizer of $\tilde{\mathcal{E}}_\sigma$.

We turn to the second orthogonal component $R_{kk}$. In full generality $R_{kk} = 0$ cannot be straightforwardly derived by considering only the gradient $\partial_{R_{kk}} \tilde{\mathcal{E}}_\sigma$ because the gradient may not be monotonous. Instead we assume that $\sigma(\chi) = 1 + \chi$. Then the gradient is

$$\partial_{R_{kk}} \tilde{\mathcal{E}}_\sigma = 2\mathbb{E}_L \mathbb{E}_{\epsilon^*, \chi^*, z^*, \xi, \zeta} \; g_\nu(\epsilon^*, \chi^*) \left( a^\top (1 + b) - z^*_{\epsilon^*} \right) a^\top \xi \tag{70}$$

$$= 2\mathbb{E}_L \mathbb{E}_{\epsilon^*, \chi^*, \xi} \; g_\nu(\epsilon^*, \chi^*) m^2_{vv^*} (1 + b)^\top \xi \tag{71}$$

$$= 2 R_{kk} m^2_{vv^*} \tag{72}$$

and consequently $R_{kk} = 0$ is the minimum of the loss.

### A.2.5 Linear attention, stability of the fixed point

We consider the case of the linear attention $\sigma(\chi) = 1 + \chi$. For this activation function we can state a more precise result about the stability of the manifold. We show that the fixed point obtained by minimizing the loss on the manifold is stable, i.e. it is a local minimum in the whole space of the order parameters $\mathbb{R}^7$.

At the fixed point obtained by minimizing the loss on the manifold we have $m_{vk^*}, m_{kv^*}, R_{vv}, R_{kk}, R_{kv} = 0, 0, 0, 0, 0$. We compute the Hessian of $\tilde{\mathcal{E}}_\sigma$ with respect to $m_{vk^*}, m_{kv^*}, R_{vv}, R_{kk}, R_{kv}$ at this point. For this we expand $\tilde{\mathcal{E}}_\sigma$ to the second order with respect to these five parameters. We have

$$\tilde{\mathcal{E}}_\sigma = \mathbb{E}_L \mathbb{E}_{\epsilon^*, \chi^*, z^*, \xi, \zeta} \; g_\nu(\epsilon^*, \chi^*) \tag{73}$$
$$\left( z^*_{\epsilon^*} - (m_{vk^*} \chi^* + m_{vv^*} z^* + R_{kv} \xi + R_{vv} \zeta)^\top \sigma(m_{kk^*} \chi^* + m_{kv^*} z^* + R_{kk} \xi + R_{kv} \zeta) \right)^2$$

$$= 1 - 2m_{vv^*} - 2(m_{vv^*} m_{kk^*} + m_{kv^*} m_{vk^*}) \mathbb{E}_L \mathbb{E}_{\epsilon^*, \chi^*} \; g_\nu(\epsilon^*, \chi^*) \chi^*_{\epsilon^*} \tag{74}$$
$$+ \mathbb{E}_L \mathbb{E}_{\epsilon^*, \chi^*} \; g_\nu(\epsilon^*, \chi^*) \Big[ \left( m_{vv^*} z^{*\top} (1 + m_{kk^*} \chi^*) \right)^2 + L(R^2_{vv} + R^2_{vk})$$
$$+ m^2_{vv^*} m^2_{kv^*} (z^{*\top} z^*)^2 + m^2_{vk^*} \left( (1 + m_{kk^*} \chi^*)^\top \chi^* \right)^2$$
$$+ m^2_{vv^*} (R^2_{vk} + R^2_{kk}) z^{*\top} z^* + m^2_{kk^*} (R^2_{vv} + R^2_{vk}) \chi^{*\top} \chi^*$$
$$+ 2m_{vv^*} m_{kv^*} m_{vk^*} z^{*\top} z^* (1 + m_{kk^*} \chi^*)^\top \chi^* + 2m_{vv^*} m_{kv^*} m_{vk^*} (1 + m_{kk^*} \chi^*)^\top z^* z^{*\top} \chi^* \Big]$$

The Hessian is diagonal and positive with respect to $R_{vv}, R_{kk}$ and $R_{kv}$; and consequently $R_{vv}, R_{kk}, R_{kv} = 0, 0, 0$ is well stable. The Hessian in the two remaining directions $m_{vk^*}$ and $m_{kv^*}$ reads

$$\partial^2_{m_{vk^*}, m_{vk^*}} \tilde{\mathcal{E}}_\sigma = \mathbb{E} \left( (1 + m_{kk^*} \chi^*)^\top \chi^* \right)^2 \tag{75}$$

$$\partial^2_{m_{kv^*}, m_{kv^*}} \tilde{\mathcal{E}}_\sigma = \mathbb{E} m^2_{vv^*} \left( z^{*\top} z^* \right)^2 \tag{76}$$

$$\partial^2_{m_{kv^*}, m_{vk^*}} \tilde{\mathcal{E}}_\sigma = 2\mathbb{E} \left[ -\chi^*_{\epsilon^*} + m_{vv^*} z^{*\top} z^* (1 + m_{kk^*} \chi^*)^\top \chi^* + m_{vv^*} (1 + m_{kk^*} \chi^*)^\top z^* z^{*\top} \chi^* \right] \tag{77}$$

where we used the shorthand $\mathbb{E}$ for $\mathbb{E}_L \mathbb{E}_{\epsilon^*, \chi^*, z^*} \; g_\nu(\epsilon^*, \chi^*)$. The trace of the Hessian is positive. The determinant of the Hessian is

$$\det = m^2_{vv^*} \mathbb{E} \left( (1 + m_{kk^*} \chi^*)^\top \chi^* \right)^2 \mathbb{E} \left( z^{*\top} z^* \right)^2 \tag{78}$$
$$- \left( \mathbb{E} \left[ -\chi^*_{\epsilon^*} + m_{vv^*} z^{*\top} z^* (1 + m_{kk^*} \chi^*)^\top \chi^* + m_{vv^*} (1 + m_{kk^*} \chi^*)^\top z^* z^{*\top} \chi^* \right] \right)^2$$

We evaluate this quantity at $m_{vv^*}, m_{kk^*}$ the minimizers of the loss on the manifold, for all the configurations considered in the article. We find that $\det > 0$ and consequently the Hessian at the fixed point is positive, that is to say the fixed point on the manifold is a local minimizer on $\mathbb{R}^7$.

### A.2.6 Other possible minima

We numerically show that if one does not restrict the space of the parameters to $\mathcal{M}$, $\mathcal{E}_\sigma$ can admit several minima. Yet, among them, the minimum reached on $\mathcal{M}$ is the best.

We perform a local minimization of $\tilde{\mathcal{E}}_\sigma$ starting from the initial condition $(m_{kk^*}, m_{kv^*}, m_{vk^*}, m_{vv^*}, R_{kk}, R_{kv}, R_{vv}) = (0, 1, 1, 0, 0, 0, 0)$. The intuition is that the attention can confuse and mix $k^*$ and $v^*$. For instance for an identity activation function $\sigma(\chi) = \chi$, $k$

and $v$ play symmetric roles, that we partially broke by adding a constant bias. We call *mismatched minimum* such a minimum where $m_{kv^*} \neq 0, m_{vk^*} \neq 0$ and $m_{kk^*} = 0, m_{vv^*} = 0$, and set $\beth_\sigma$ its risk, by opposition with the *well matched minimum* on $\mathcal{M}$ where $m_{kk^*} \neq 0, m_{vv^*} \neq 0$ and $m_{kv^*} = 0, m_{vk^*} = 0$. On Figure 5 we show that at large enough sequence length $L = 10$ and at $\nu > 0$, on the two models, the attention always has a well matched and a mismatched minimum, and that the mismatched minimum has a larger population error $\beth_\sigma > \mathsf{E}_\sigma$.

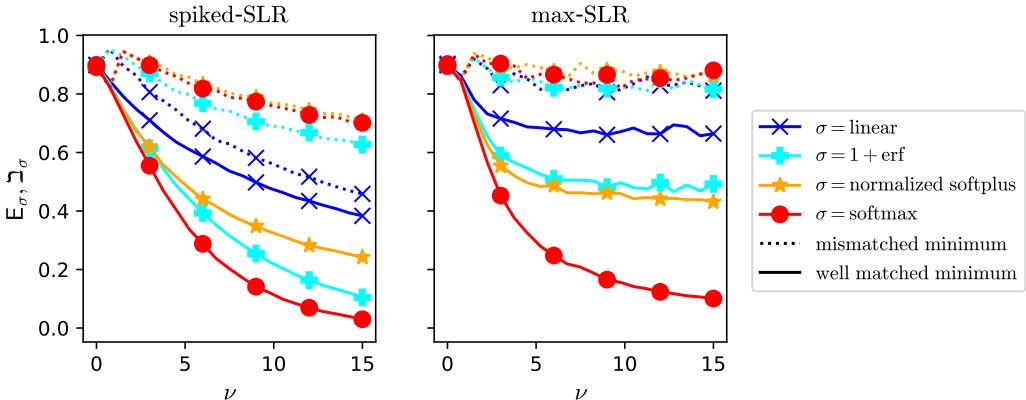

Figure 5: Minimal population risk $\mathsf{E}_\sigma$, reached at the well matched minimum, and population risk of the mismatched minimum $\beth_\sigma$, for different attention activations $\sigma$ (colors), for the two tasks spiked-SLR and max-SLR at $L = 10$. The markers on the lines are for readability only. Population risks are computed by numerical optimization of $\tilde{\mathcal{E}}_\sigma$ from a random initialization (well matched minimum) or from the mismatched initialization described in section A.2.6 (mismatched minimum).

### A.2.7 Minimizer for the linear attention

On the manifold $\mathcal{M}$ the loss can be analytically minimized. We state here the expression of the minimizer. The loss is

$$
\begin{aligned}
\tilde{\mathcal{E}}_\sigma &= \mathbb{E}_L \mathbb{E}_{\epsilon^*, \chi^*, z^*, \xi, \zeta} \; g_\nu(\epsilon^*, \chi^*) \left( z^*_{\epsilon^*} - (m_{vv^*} z^* + R_{vv}\zeta)^\top (1 + m_{kk^*}\chi^* + R_{kk}\xi) \right)^2 \\
&= 1 - 2m_{vv^*} - 2m_{vv^*} m_{kk^*} \mathbb{E}_L \mathbb{E}_{\epsilon^*, \chi^*} \; g_\nu(\epsilon^*, \chi^*) \chi^*_{\epsilon^*} \\
&\quad + \mathbb{E}_L \mathbb{E}_{\epsilon^*, \chi^*} \; g_\nu(\epsilon^*, \chi^*) \left[ m^2_{vv^*} (1 + m_{kk^*}\chi^*)^\top (1 + m_{kk^*}\chi^*) \right. \\
&\quad \left. + LR^2_{vv} + Lm^2_{vv^*} R^2_{kk} + m^2_{kk^*} R^2_{vv} \chi^{*\top}\chi^* \right]
\end{aligned}
\tag{79}
$$

At the minimizer $R_{vv} = 0$ and $R_{kk} = 0$. We perform the minimization over the remaining variables and obtain that the loss is

$$
\tilde{\mathcal{E}}_\sigma = 1 - \frac{(\mathbb{E}(1 + m_{kk^*}\chi^*)_{\epsilon^*})^2}{\mathbb{E}(1 + m_{kk^*}\chi^*)^\top (1 + m_{kk^*}\chi^*)} \qquad m_{kk^*} = \frac{\mathbb{E}L\mathbb{E}\chi^*_{\epsilon^*} - \mathbb{E}1^\top \chi^*}{\mathbb{E}\chi^{*\top}\chi^* - \mathbb{E}\chi^*_{\epsilon^*}\mathbb{E}1^\top \chi^*}
\tag{80}
$$

where we used the shorthand $\mathbb{E}$ for $\mathbb{E}_L \mathbb{E}_{\epsilon^*, \chi^*} \; g_\nu(\epsilon^*, \chi^*)$. We evaluate these quantities for the two models, the spiked- and the max-SLR. We set $f(L, \nu) = \mathbb{E}_{\epsilon^*, \chi^*} \; g_\nu(\epsilon^*, \chi^*) \chi^*_{\epsilon^*}$, which for the max-SLR has no simpler expression in general. We compute:

|  | $\mathbb{E}\chi^*_{\epsilon^*}$ | $\mathbb{E}1^\top \chi^*$ | $\mathbb{E}\chi^{*\top}\chi^*$ |
|---|---|---|---|
| spiked $-$ SLR | $\sqrt{\nu}$ | $\sqrt{\nu}$ | $\mathbb{E}L + \nu$ |
| max $-$ SLR | $\mathbb{E}f(L, \nu)$ | $0$ | $\mathbb{E}L$ |

The optimal risk for the spiked-SLR finally is

$$
\mathsf{E}_\sigma = 1 - \frac{\mathbb{E}L + \nu(\mathbb{E}L - 1)}{(\mathbb{E}L)^2 + \nu(\mathbb{E}L - 1)}
\tag{81}
$$

while for the max-SLR the optimal risk is

$$
\mathsf{E}_\sigma = 1 - \frac{1 + (\mathbb{E}f(L, \nu))^2}{\mathbb{E}L} \, .
\tag{82}
$$

### A.2.8 PROOF OF COROLLARY 4.3: ASYMPTOTIC CONVERGENCE RATES

**Spiked-SLR at large signal $\nu \to \infty$ and constant length $L$.** The loss of the linear attention is derived in part A.2.7. We take the limit $\nu \to \infty$:

$$\mathsf{E}_\sigma = 1 - \frac{L + \nu(L-1)}{L^2 + \nu(L-1)} = \frac{L}{L-1}\frac{1}{\nu} + o_{\nu \to \infty}(1) \tag{83}$$

The loss of the softmax attention is derived in part A.2.1. For the spiked-SLR it reads

$$\mathsf{E}_\sigma = 1 - \mathbb{E}_\chi \frac{1}{L} \sum_\epsilon^L \frac{g(\epsilon,\chi)^2}{\sum_{\epsilon'}^L g(\epsilon',\chi)} \tag{84}$$

$$= 1 - \mathbb{E}_{\chi_1 \sim \mathcal{N}(\sqrt{\nu},1)} \prod_{l>1}^L \left[\mathbb{E}_{\chi_l \sim \mathcal{N}(0,1)}\right] \frac{e^{\sqrt{\nu}\chi_1}}{\sum_{l'=1}^L e^{\sqrt{\nu}\chi_{l'}}} \tag{85}$$

where we isolated the first index by symmetry and made a change of variable on $\chi_1$. At large $\nu$ we approximate the softmax by a hardmax and

$$\mathsf{E}_\sigma \approx 1 - \mathbb{E}_{\chi_1 \sim \mathcal{N}(\sqrt{\nu},1)} \prod_{l>1}^L \left[\mathbb{E}_{\chi_l \sim \mathcal{N}(0,1)}\right] \delta_{\chi_1 > \max_{l>1}\chi_l} \tag{86}$$

$$= \prod_{l>1}^L \left[\mathbb{E}_{\chi_l \sim \mathcal{N}(0,1)}\right] e^{-\frac{1}{2}(\nu - \max_{l>1}\chi_l) + o_{\nu \to \infty}(\nu)} \tag{87}$$

$$= e^{-\frac{1}{2}\nu + o_{\nu \to \infty}(\nu)} \tag{88}$$

**Max-SLR at $\nu = \infty$ for growing lengths $L$.** We take the limit $\nu = \infty$ so $g_\nu(\epsilon,\chi) = L\delta_{\epsilon = \arg\max_l \chi_l}$ and $f(L,\nu) = \mathbb{E}_{\chi^* \sim (0,I_L)} \max_l \chi_l^*$. We assume constant length, i.e. $P_L$ is a delta. The loss of the linear attention is derived in part A.2.7:

$$\mathsf{E}_\sigma = 1 - \frac{1 + (\mathbb{E}f(L,\nu))^2}{\mathbb{E}L} . \tag{89}$$

For large $L$ we have the asymptotic $f(L,\nu) = \mathcal{O}_{L \to \infty}(\sqrt{\log L})$, which gives

$$\mathsf{E}_\sigma = 1 - \mathcal{O}_{L \to \infty}\left(\frac{\log L}{L}\right) . \tag{90}$$

The loss of the softmax attention is derived in part A.2.1. For the max-SLR it reads

$$\mathsf{E}_\sigma = 1 - \mathbb{E}_\chi \frac{1}{L} \sum_\epsilon^L \frac{g(\epsilon,\chi)^2}{\sum_{\epsilon'}^L g(\epsilon',\chi)} \tag{91}$$

$$= 1 - \mathbb{E}_\chi \frac{1}{L} \sum_\epsilon^L g(\epsilon,\chi) \tag{92}$$

$$= 0 \tag{93}$$

The softmax attention reaches exact recovery for any $L$, and in particular for large $L$, the limit been taken after the limit $\nu = \infty$.

## B BAYES-OPTIMAL ERROR AT FINITE SAMPLE COMPLEXITY $\alpha$

In this section we state the expression for the Bayes-optimal test error in the case where samples are limited, i.e. when $\alpha = {}^N/_D$ is finite, in the high-dimensional limit $N, D \to \infty$. The derivation of this expression is given in appendix C. We analyze behaviour of the BO performances and show that the SLR task presents a hard phase where best algorithmic performances cannot reach best information-theoretic performances. This hard phase is not present at $\alpha \to \infty$.

The Bayes-optimal performances can be expressed in function of two low-dimensional order parameters (or summary statistics, or sufficient statistics), $m_k \in \mathbb{R}$ and $m_v \in \mathbb{R}$. Writing $k_{\mathrm{BO}} \in \mathbb{R}^D$ and $v_{\mathrm{BO}} \in \mathbb{R}^D$ the BO estimators of $k^*$ and $v^*$, $m_k$ and $m_v$ are defined as

$$m_k = \frac{1}{D} k_{\mathrm{BO}}^\top k^* = \mathrm{angle}(k_{\mathrm{BO}}, k^*)^2 \,, \qquad m_v = \frac{1}{D} v_{\mathrm{BO}}^\top v^* = \mathrm{angle}(v_{\mathrm{BO}}, v^*)^2 \,. \qquad (94)$$

To state our asymptotic characterization result we introduce the following partition functions, for all $L$ and for $B, y \in \mathbb{R}$, $A, R, V \in \mathbb{R}^+$ and $\gamma, \omega \in \mathbb{R}^L$:

$$Z_k(B, A) = \mathbb{E}_{k \sim \mathcal{N}(0,1)} e^{-\frac{1}{2} A k^2 + B k} \,, \qquad Z_v(B, A) = \mathbb{E}_{v \sim \mathcal{N}(0,1)} e^{-\frac{1}{2} A v^2 + B v} \qquad (95)$$

$$Z_{\mathrm{out}}(y, \gamma, R, \omega, V) = \int_{\mathbb{R}^L} \mathrm{d}\chi \mathrm{d}z \frac{1}{L} \sum_\epsilon^L g_\nu(\epsilon, \chi) P_{\mathrm{out}}(y|z_\epsilon) \prod_l^L \mathcal{N}(\chi_l; \gamma_l, R) \mathcal{N}(z_l; \omega_l, V) \qquad (96)$$

where

$$P_{\mathrm{out}}(y|z_{\epsilon^*}^*) = \frac{e^{-\frac{1}{2\Delta}(y - z_{\epsilon^*}^*)^2}}{\sqrt{2\pi\Delta}}$$

is the output channel that corresponds to an additive Gaussian noise. We set $h_\nu$ an effective distribution on $\epsilon \in \{1, \ldots, L\}$ defined as

$$h_\nu(\epsilon, \gamma, R) = \int_{\mathbb{R}^L} \mathrm{d}\chi \, g_\nu(\epsilon, \chi) \prod_l^L \mathcal{N}(\chi_l; \gamma_l, R) \,. \qquad (97)$$

Last we set the function xlogx : $x \to x \log(x)$.

**Result B.1** (Bayes-optimal risk). *We consider the high-dimensional limit $N, D \to \infty$, with $\alpha = \Omega(1)$. Let $L \sim P_L$ and, conditionally on $L$, $\varsigma \sim \mathcal{N}(0,1)$, $\xi \sim \mathcal{N}(0, I_L)$ and $\zeta \sim \mathcal{N}(0, I_L)$. Fix $m_k$ and $m_v$ so they satisfy the following fixed-point condition:*

$$m_k, m_v = \underset{m_k, m_v}{\arg\max} \, \phi_{\mathrm{BO}}(m_k, m_v) \qquad (98)$$

$$\phi_{\mathrm{BO}}(m_k, m_v) = \max_{\hat{m}_k \in \mathbb{R}, \hat{m}_v \in \mathbb{R}} \left[ -\frac{1}{2\alpha}(\hat{m}_k m_k + \hat{m}_v m_v) + \frac{1}{\alpha} \mathbb{E}_\varsigma \mathrm{xlogx} Z_k \left( \sqrt{\hat{m}_k} \varsigma, \hat{m}_k \right) \right. \qquad (99)$$

$$\left. + \frac{1}{\alpha} \mathbb{E}_\varsigma \mathrm{xlogx} Z_v \left( \sqrt{\hat{m}_v} \varsigma, \hat{m}_v \right) + \mathbb{E}_L \mathbb{E}_{\xi, \zeta} \int_{\mathbb{R}} \mathrm{d}y \, \mathrm{xlogx} Z_{\mathrm{out}} \left( y, \sqrt{m_k} \xi, 1 - m_k, \sqrt{m_v} \zeta, 1 - m_v \right) \right]$$

*$\phi_{\mathrm{BO}}$ is the free entropy of the problem at given $m_k$ and $m_v$. Then the Bayes-optimal (BO) test error on inferring $y$ is:*

$$\mathcal{E}_{\mathrm{BO}} = 1 - m_v \mathbb{E}_L \mathbb{E}_\xi \frac{1}{L} \left[ \sum_\epsilon^L h_\nu(\epsilon, \sqrt{m_k} \xi, 1 - m_k) \right]^{-1} \sum_\epsilon^L h_\nu(\epsilon, \sqrt{m_k} \xi, 1 - m_k)^2 \,. \qquad (100)$$

Our above result describes the information-theoretical (IT) best performance. In general it may not correspond to the algorithmic best performance. In fact, the SLR task presents an interesting phenomenology: in a whole region of the parameters $\alpha$, $\nu$, $L$ of the model, there is a gap between the IT and algorithmic achievable best performances. Such a discrepancy is related to the existence of several maxima in the free entropy $\phi_{\mathrm{BO}}$ eq. (99) and has already been studied for other models in numerous previous works (Krzakala et al., 2012; Dia et al., 2016; Lesieur et al., 2017) and we refer to Bandeira et al. (2022) for a more rigorous treatment. The free entropy $\phi_{\mathrm{BO}}$ plays a central role and it is related to the log-likelihood of the model; higher free entropy gives lower risk.

We can show that $\phi_{\mathrm{BO}}$ admits one or two maxima. We introduce some notation to distinguish them. At given $\nu, L$ we call $\alpha_{\mathrm{algo}}$ the *algorithmic threshold* (or spinodal) the smallest $\alpha$ such that for all $\alpha > \alpha_{\mathrm{alg}}$, $\phi_{\mathrm{BO}}$ has a unique maximum. In the region $\alpha < \alpha_{\mathrm{alg}}$ where $\phi_{\mathrm{BO}}$ admits two maxima, we set $\phi_{\mathrm{BO}}^i$ the maximum whose basin of attraction includes a neighborhood of $m_k, m_v = 1, 1$ and call it *informed maximum*. We set $\phi_{\mathrm{BO}}^u$ the maximum of $\phi_{\mathrm{BO}}$ whose basin of attraction includes a neighborhood of $m_k, m_v = 0, 0$ and call it *uninformed maximum*; it describes the algorithmic best performances, in the sense that an algorithm locally optimizing $\phi_{\mathrm{BO}}$ and starting without information on the solution will reach $\phi_{\mathrm{BO}}^u$. In general $\phi_{\mathrm{BO}}^u \neq \phi_{\mathrm{BO}}^i$. We call $\alpha_{\mathrm{IT}}$ the *IT threshold* the $\alpha$ such

that $\phi_{\mathrm{BO}}^{\mathrm{i}} = \phi_{\mathrm{BO}}^{\mathrm{u}}$. We have $\alpha_{\mathrm{IT}} \le \alpha_{\mathrm{alg}}$ and we compute that for the SLR $\alpha_{\mathrm{IT}} = 1$ for a noise-less output channel $\Delta = 0$.

To summarize, in the region $\alpha_{\mathrm{IT}} < \alpha < \alpha_{\mathrm{alg}}$ the free entropy $\phi_{\mathrm{BO}}$ has two maxima; the global maximum describes the IT best performances but it cannot be reached from $m_k, m_v = 0, 0$ by local algorithms, that only reach a local maximum with higher risk. We depict this hard phase on Figure 6 for the spiked- and max-SLR.

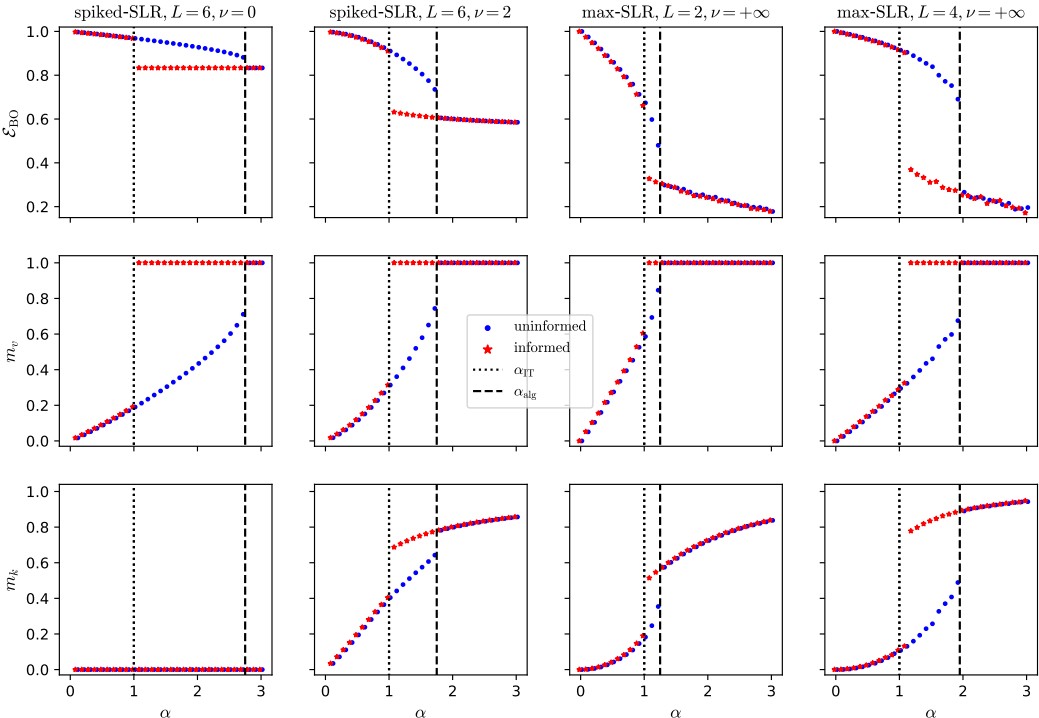

Figure 6: BO test error $\mathcal{E}_{\mathrm{BO}}$, overlap $m_v$ with $v^*$ and overlap $m_k$ with $k^*$ for the spiked-SLR and the max-SLR models, for different signals $\nu$ and sequence lengths $L$, in the noise-less case $\Delta = 0$. The values are given by the eq. (100). The algorithmic best performances are given by the "uninformed" curve while for $\alpha_{\mathrm{IT}} < \alpha < \alpha_{\mathrm{alg}}$ the IT best performances are given by the "informed" curve. For $\alpha < \alpha_{\mathrm{IT}}$ and $\alpha_{\mathrm{alg}} < \alpha$ the two curves are equal.

Considering Figure 6 we can state that the extent of the hard phase $\alpha_{\mathrm{alg}} - \alpha_{\mathrm{IT}}$ decreases with the signal $\nu$ and grows with the sequence length $L$. At finite $L$ the upper limit of the hard phase $\alpha_{\mathrm{alg}}$ is finite and consequently on population loss at $\alpha = +\infty$ the hard phase is not sensible. Notice that since we consider a noise-less setting $v^*$ can be exactly inferred at finite $\alpha$.

A possible explanation of why such a hard phase exists is that it is induced by the sparsity of the SLR. If one flatten each sequence into a single token $\tilde{X}$ of dimension $LD$ one would have a setting similar to compress sensing where $y = \tilde{X}^\top \tilde{v}^*$, where the regression vector $\tilde{v}^* \in \mathbb{R}^{LD}$ has a proportion $(L-1)/L$ of null entries. Compress sensing has been shown to present a similar hard phase (Krzakala et al., 2012). More generally sparsity induces hard phase, as shown for sparse PCA (Dia et al., 2016) or strongly unbalanced binary SBM (Caltagirone et al., 2016).

An interesting particular case is the 0-SLR when $\nu = 0$ and the tokens are iid Gaussians bringing no information alone. As shown on Figure 6 it is still possible to exactly recover $v^*$ and perform better than random. Indeed one can compare the train label $y$ to $X_l^\top v$ for each token $l$. If $v$ has some correlations with $v^*$ then $(y - X_l^\top v)^2$ is likely to be smaller for $l = \epsilon^*$. This gives some information on the probable relevant token. Then one can refine $v$ by focusing more on the right $X_l$. At inference time one cannot use the label to guess $\epsilon^*$ and the prediction is simply $\sum_l X_l^T v$.

## C  Asymptotic characterization at finite sampling ratio: replica

In this part we derive the asymptotic characterization of the Bayes-optimal performances and the performances of the trained attention, at finite sampling ratio $\alpha$. We jointly treat the two cases, the BO and the attention, and the two models, spiked-SLR and max-SLR, taking a variable sequence length. The derivation is done for a general output channel $P_{\text{out}}$ such that for each sample the label $y$ is distributed with density $P_{\text{out}}(y|z_{\epsilon*}^*)$. The additive Gaussian noise channel considered in the main part then corresponds to

$$P_{\text{out}}(y|z_{\epsilon*}^*) = \frac{e^{-\frac{1}{2\Delta^2}(y-z_{\epsilon*}^*)^2}}{\sqrt{2\pi\Delta^2}} . \tag{101}$$

Moreover the derivation is done for any convex loss function with strictly convex regularization.

### C.1  Introduction of the partition function

We consider data $X, y$ made of $N$ train samples, indexed by $\mu = 1, \ldots, N$, together with $N' = \rho N$ test samples, with $\rho > 0$, indexed by $\mu = N + 1, \ldots, N + N'$. They are distributed according to the generative model defined in the main part 3.1. We will take the limit $\rho \to 0$ at the end of the derivation, so no information can be extracted from the test samples $X_{N+1 \leq \mu \leq N+N'}$ in a unsupervised way by the Bayes-optimal estimator.

The expected test square error of the attention-based estimator is

$$\mathsf{E}_\sigma(\alpha) = \mathbb{E}_{X,y} \frac{1}{N'} \sum_{\mu=N+1}^{N+N'} (y_\mu - f_{\sigma,k,v}(X_\mu))^2 \tag{102}$$

where

$$f_{\sigma,k,v}(X_\mu) = \sigma\left(\frac{1}{\sqrt{D}}X_\mu k\right)^\top \frac{1}{\sqrt{D}}X_\mu v \tag{103}$$

$$k, v = \arg\min_{k,v} \mathcal{L}(k,v) \tag{104}$$

$$\mathcal{L}(k,v) = \sum_{\mu=1}^{N} \ell(y_\mu, f_{\sigma,k,v}(X_\mu)) + r_k \sum_{i}^{D} \gamma(k_i) + r_v \sum_{i}^{D} \gamma(v_i) \tag{105}$$

with $\ell(x, x') = \frac{1}{2}(x - x')^2$ the square loss and $\gamma(x) = \frac{1}{2}x^2$ the $l_2$ regularization. Notice that the following derivation is done in full generality for any convex $\ell$ and strictly convex $\gamma$. We will specialize to ridge regression at the end of the derivation.

The expected test square error of the BO estimator is

$$\mathcal{E}_{\text{BO}} = \mathbb{E}_{X,y} \frac{1}{N'} \sum_{\mu=N+1}^{N+N'} (y_\mu - \hat{y}_\mu^{\text{BO}})^2 = \mathbb{E}_{X,y} \frac{1}{N'} \sum_{\mu=N+1}^{N+N'} \left(y_\mu^2 - 2y_\mu \hat{y}_\mu^{\text{BO}} + (\hat{y}_\mu^{\text{BO}})^2\right) \tag{106}$$

where $\hat{y}^{\text{BO}}$ is the optimal estimator in terms of expected square error, that is, for a test sample $\mu' > N$:

$$\hat{y}_{\mu'}^{\text{BO}} = \int \mathrm{d}y_{\mu'}\, y_{\mu'} P(y_{\mu'}|X, y_{\mu \leq N}) \tag{107}$$

Notice that, since $\hat{y}_{\mu'}^{\text{BO}}$ is sampled from the posterior distribution, we have the simplification $\mathbb{E}_{X,y} (\hat{y}_{\mu'}^{\text{BO}})^2 = \mathbb{E}_{X,y} y_{\mu'} \hat{y}_{\mu'}^{\text{BO}}$. We can expand the posterior distribution over the test labels as

$$P(y_{\mu>N}|X, y_{\mu \leq N}) = \int \mathrm{d}\epsilon \mathrm{d}k \mathrm{d}v\, P(y_{\mu>N}, \epsilon, k, v|X, y_{\mu \leq N}) \tag{108}$$

$$\propto \int \mathrm{d}P(\epsilon, k, v|X)\, P(y|X, \epsilon, k, v) \tag{109}$$

$$\propto \int \mathrm{d}P_k(k)\mathrm{d}P_v(v) \prod_{\mu=1}^{N+N'} \mathrm{d}P_\epsilon(\epsilon_\mu)\, g_\nu\left(\epsilon_\mu, \frac{1}{\sqrt{D}}X_\mu k\right) P\left(y_\mu|\frac{1}{\sqrt{D}}(X_\mu v)_{\epsilon_\mu}\right) \tag{110}$$

with $P_k = P_v = \mathcal{N}(0, I_D)$ the prior distribution on $k$ and $v$ and, with an abuse of notation, $P_\epsilon = \mathrm{Unif}(\{1, \ldots, L(\mu)\})$ the prior on $\epsilon_\mu$. We used Bayes rule and $P(X|\epsilon, k, v) \propto g_\nu(\epsilon, Xk/\sqrt{D})P(\epsilon)$ to rewrite $P(\epsilon, k, v|X)$; and we used the independence of the samples $X_\mu, y_\mu$ conditional on $\epsilon_\mu$, $k$ and $v$ to factorize the expression.

We can express the two errors $\mathsf{E}_\sigma(\alpha)$ and $\mathcal{E}_{\mathrm{BO}}$ under the same formalism thanks to a partition function and a free entropy over $k$ and $v$:

$$Z(X, y) = \int \mathrm{d}\tilde{P}_k(k)\mathrm{d}\tilde{P}_v(v) \int \prod_{\mu=1}^{N} \mathrm{d}P_\epsilon(\epsilon_\mu)\, \tilde{g}_\nu\left(\epsilon_\mu, \frac{1}{\sqrt{D}}X_\mu k\right) \tilde{P}(y_\mu|X_\mu, \epsilon_\mu, k, v) \tag{111}$$

$$\int \prod_{\mu=N+1}^{N+N'} \mathrm{d}\hat{y}_\mu \mathrm{d}P_\epsilon(\epsilon_\mu)\, \tilde{g}_\nu\left(\epsilon_\mu, \frac{1}{\sqrt{D}}X_\mu k\right) \hat{P}(\hat{y}_\mu|X_\mu, \epsilon_\mu, k, v)\, e^{\sum_{\mu=N+1}^{N+N'}\left(sy_\mu(y_\mu-\hat{y}_\mu)+t\beta(y_\mu-\hat{y}_\mu)^2\right)}$$

$$\phi = \frac{1}{N}\mathbb{E}_{X,y} \log Z(X, y) \tag{112}$$

where, depending on the estimator, the densities are taken according to

| | $\tilde{P}_k(k)$ | $\tilde{P}_v(v)$ | $\tilde{g}_\nu$ | $\tilde{P}(y_\mu|X_\mu, \epsilon_\mu, k, v)$ | $\hat{P}(\hat{y}_\mu|X_\mu, \epsilon_\mu, k, v)$ |
|---|---|---|---|---|---|
| attention | $e^{-\beta r_k \sum_i^D \gamma(k_i)}$ | $e^{-\beta r_v \sum_i^D \gamma(v_i)}$ | $1$ | $e^{-\beta\ell(y_\mu, f_{\sigma,k,v}(X_\mu))}$ | $\delta(\hat{y}_\mu - f_{\sigma,k,v}(X_\mu))$ |
| BO | $P_k(k)$ | $P_v(v)$ | $g_\nu$ | $P(y_\mu|X_\mu, \epsilon_\mu, v)$ | $P(\hat{y}_\mu|X_\mu, \epsilon_\mu, v)$ |

with $\beta \to \infty$, so for the attention $k$ and $v$ concentrate onto the minimizer of the empirical loss. We introduced tilting variables $s$ and $t$ to access the expected test errors, obtained according to

$$\mathsf{E}_\sigma(\alpha) = \lim_{\beta\to\infty} \frac{\partial}{\partial t}\frac{1}{\rho\beta}\phi|_{s=0,t=0}\,, \tag{113}$$

$$\mathcal{E}_{\mathrm{BO}} = \frac{\partial}{\partial s}\frac{1}{\rho}\phi|_{s=0,t=0}\,. \tag{114}$$

## C.2 Replica, free entropy

We compute the free entropy $\phi$ in the high-dimensional limit $N, D \to \infty$. We use the replica trick:

$$\mathbb{E}_{X,y} \log Z(X, y) = \partial_n(\mathbb{E}_{X,y} Z(X, y)^n)|_{n=0} \tag{115}$$

We introduce $n$ replica, indexed by $a = 1, \ldots, n$, and count the expectation over the data $X, y$ as an additional replica indexed by $a = 0$ corresponding to the ground truth. We equivalently use the superscript $*$ to denote the ground truth when it is clearer to do so. We use the shorthand $O_\mu^a = sy_\mu(y_\mu - \hat{y}_\mu^a) + t\beta(y_\mu - \hat{y}_\mu^a)^2$ for the observables. This gives

$$\mathbb{E}_{X,y}Z^n \propto \mathbb{E}_{X,y} \int \prod_{a=1}^{n} \mathrm{d}\tilde{P}_k(k^a)\mathrm{d}\tilde{P}_v(v^a) \prod_{\mu}^{N+N'} \prod_{a=1}^{n} \mathrm{d}P_\epsilon(\epsilon_\mu^a)\, \tilde{g}_\nu\left(\epsilon_\mu^a, \frac{1}{\sqrt{D}}X_\mu k^a\right) \tag{116}$$

$$\prod_{\mu}^{N} \prod_{a=1}^{n} \tilde{P}^a(y_\mu|X_\mu, \epsilon_\mu^a, k^a, v^a) \prod_{\mu>N} \prod_{a=1}^{n} \mathrm{d}\hat{y}_\mu^a \hat{P}(\hat{y}_\mu^a|X_\mu, \epsilon_\mu^a, k^a, v^a)\, e^{O_\mu^a}$$

$$\propto \mathbb{E}_{X\sim\mathcal{N}} \int \prod_{a=0}^{n} \mathrm{d}\tilde{P}_k^a(k^a)\mathrm{d}\tilde{P}_v^a(v^a) \prod_{\mu}^{N+N'} \prod_{a=0}^{n} \mathrm{d}P_\epsilon(\epsilon_\mu^a)\, \tilde{g}_\nu^a\left(\epsilon_\mu^a, \frac{1}{\sqrt{D}}X_\mu k^a\right) \tag{117}$$

$$\prod_{\mu}^{N} \mathrm{d}y_\mu \prod_{a=0}^{n} \tilde{P}^a(y_\mu|X_\mu, \epsilon_\mu^a, k^a, v^a) \prod_{\mu>N} \mathrm{d}y_\mu P(y_\mu|X_\mu, \epsilon_\mu^*, v^*) \prod_{a=1}^{n} \mathrm{d}\hat{y}_\mu^a \hat{P}(\hat{y}_\mu^a|X_\mu, \epsilon_\mu^a, k^a, v^a)\, e^{O_\mu^a}$$

where $\tilde{P}_k^a, \tilde{P}_v^a, \tilde{g}_\nu^a, \tilde{P}^a$ equals $P_k, P_v, g_\nu, P$ if $a = 0$ and $\tilde{P}_k, \tilde{P}_v, \tilde{g}_\nu, \tilde{P}$ otherwise. We now introduce the two projections

$$\chi_\mu^a = \frac{1}{\sqrt{D}}X_\mu k^a \in \mathbb{R}^{L(\mu)}\,, \qquad z_\mu^a = \frac{1}{\sqrt{D}}X_\mu v^a \in \mathbb{R}^{L(\mu)} \tag{118}$$

thanks to Dirac deltas. We integrate over Gaussian $X$. We pack the replica into vectors of size $n+1$. This gives

$$\mathbb{E}_{X,y} Z^n \propto \mathbb{E}_{X \sim \mathcal{N}} \int \prod_{a=0}^{n} \mathrm{d}\tilde{P}_k^a(k^a) \mathrm{d}\tilde{P}_v^a(v^a) \prod_{\mu}^{N+N'} \prod_{a=0}^{n} \mathrm{d}\chi_\mu^a \mathrm{d}\hat{\chi}_\mu^a \mathrm{d}z_\mu^a \mathrm{d}\hat{z}_\mu^a dP_\epsilon(\epsilon_\mu^a)\, \tilde{g}_\nu^a\left(\epsilon_\mu^a, \chi_\mu^a\right) \quad (119)$$

$$\prod_{\mu}^{N} \mathrm{d}y_\mu \prod_{a=0}^{n} \tilde{P}^a(y_\mu | z_\mu^a, \chi_\mu^a, \epsilon_\mu^a) \prod_{\mu>N} \mathrm{d}y_\mu P(y_\mu | z_\mu^*, \epsilon_\mu^*) \prod_{a=1}^{n} \mathrm{d}\hat{y}_\mu^a \hat{P}(\hat{y}_\mu^a | z_\mu^a, \chi_\mu^a, \epsilon_\mu^a) e^{O_\mu^a}$$

$$\prod_{a=0}^{n} \prod_{\mu}^{N+N'} e^{\sum_l^{L(\mu)} \mathrm{i}\hat{\chi}_{\mu,l}^a \left(\chi_{\mu,l}^a - \frac{1}{\sqrt{D}} X_{\mu,l} k^a\right) + \mathrm{i}\hat{z}_{\mu,l}^a \left(z_{\mu,l}^a - \frac{1}{\sqrt{D}} X_{\mu,l} v^a\right)}$$

$$\propto \int \prod_{a=0}^{n} \mathrm{d}\tilde{P}_k^a(k^a) \mathrm{d}\tilde{P}_v^a(v^a) \prod_{\mu}^{N+N'} \mathrm{d}P_L(L(\mu)) \prod_{a=0}^{n} \mathrm{d}\chi_\mu^a \mathrm{d}z_\mu^a dP_\epsilon(\epsilon_\mu^a)\, \tilde{g}_\nu^a\left(\epsilon_\mu^a, \chi_\mu^a\right) \quad (120)$$

$$\prod_{\mu}^{N} \mathrm{d}y_\mu \prod_{a=0}^{n} \tilde{P}^a(y_\mu | z_\mu^a, \chi_\mu^a, \epsilon_\mu^a) \prod_{\mu>N} \mathrm{d}y_\mu P(y_\mu | z_\mu^*, \epsilon_\mu^*) \prod_{a=1}^{n} \mathrm{d}\hat{y}_\mu^a \hat{P}(\hat{y}_\mu^a | z_\mu^a, \chi_\mu^a, \epsilon_\mu^a) e^{O_\mu^a}$$

$$\prod_{\mu}^{N+N'} \prod_{l}^{L(\mu)} \mathcal{N}\left( \begin{pmatrix} \chi_{\mu,l} \\ z_{\mu,l} \end{pmatrix}; 0, \begin{pmatrix} 1 & \underline{m}_{kk^*}^\top & 0 & \underline{m}_{vk^*}^\top \\ \underline{m}_{kk^*} & \underline{Q}_{kk} & \underline{m}_{kv^*}^\top & \underline{Q}_{kv} \\ 0 & \underline{m}_{kv^*} & 1 & \underline{m}_{vv^*}^\top \\ \underline{m}_{vk^*} & \underline{Q}_{kv}^\top & \underline{m}_{vv^*}^\top & \underline{Q}_{vv} \end{pmatrix} \right)$$

where we used that $(k^*)^\top k^*/D = (v^*)^\top v^*/D = 1$, $(k^*)^\top v^*/D = 0$ and we introduced the overlaps $\underline{m}_{kk^*}, \underline{m}_{vk^*}, \underline{m}_{kv^*}, \underline{m}_{vv^*} \in \mathbb{R}^n$ and $\underline{Q}_{kk}, \underline{Q}_{kv}, \underline{Q}_{vv} \in \mathbb{R}^{n \times n}$ defined for $a, b > 0$ by

$$(\underline{m}_{kk^*})_a = \frac{1}{D}(k^*)^\top k^a\,, \quad (\underline{m}_{vk^*})_a = \frac{1}{D}(k^*)^\top v^a\,, \quad (\underline{m}_{kv^*})_a = \frac{1}{D}(v^*)^\top k^a\,, \quad (\underline{m}_{vv^*})_a = \frac{1}{D}(v^*)^\top v^a$$
$$(121)$$

$$(\underline{Q}_{kk})_{ab} = \frac{1}{D}(k^a)^\top k^b\,, \quad (\underline{Q}_{kv})_{ab} = \frac{1}{D}(v^a)^\top k^b\,, \quad (\underline{Q}_{vv})_{ab} = \frac{1}{D}(v^a)^\top v^b \quad (122)$$

We introduce these overlaps via new Dirac deltas. We leverage the replica-symmetric assumption: we assume that there are $m_{kk^*}, m_{vk^*}, m_{kv^*}, m_{vv^*}, q_{kk}, q_{kv}, q_{vv}, Q_{kk}, Q_{kv}, Q_{vv} \in \mathbb{R}$ such that for all $a$ and $b$

$$(\underline{m}_{kk^*})_a = m_{kk^*}\,, \quad (\underline{m}_{vk^*})_a = m_{vk^*}\,, \quad (\underline{m}_{kv^*})_a = m_{kv^*}\,, \quad (\underline{m}_{vv^*})_a = m_{vv^*} \quad (123)$$
$$(\underline{Q}_{kk})_{ab} = q_{kk}\delta_{a\neq b} + Q_{kk}\delta_{a=b}\,, \quad (\underline{Q}_{kv})_{ab} = q_{kv}\delta_{a\neq b} + Q_{kv}\delta_{a=b}\,, \quad (\underline{Q}_{vv})_{ab} = q_{vv}\delta_{a\neq b} + Q_{vv}\delta_{a=b} \quad (124)$$

We pack these values into matrices

$$m = \begin{pmatrix} m_{kk^*} & m_{kv^*} \\ m_{vk^*} & m_{vv^*} \end{pmatrix} \in \mathbb{R}^{2\times 2}\,, \quad q = \begin{pmatrix} q_{kk} & q_{kv} \\ q_{kv} & q_{vv} \end{pmatrix} \in \mathbb{R}^{2\times 2}\,, \quad Q = \begin{pmatrix} Q_{kk} & Q_{kv} \\ Q_{kv} & Q_{vv} \end{pmatrix} \in \mathbb{R}^{2\times 2} \quad (125)$$

We factorize the replica by introducing random Gaussian variables $\varsigma \sim \mathcal{N}(0, I_2)$, $\xi \sim \mathcal{N}(0, I_L)$ and $\zeta \sim \mathcal{N}(0, I_L)$. After a standard computation (see e.g. Aubin et al. (2020)) we obtain that the free entropy can be expressed as an extremum over the order parameters $\Theta = (m, q, Q)$ and their conjugates $\hat{\Theta} = (\hat{m} \in \mathbb{R}^{2\times 2}, \hat{q} \in \mathbb{R}^{2\times 2}, \hat{Q} \in \mathbb{R}^{2\times 2})$:

$$\phi = \max_{\Theta, \hat{\Theta}} \phi(\Theta, \hat{\Theta}) \quad (126)$$

$$\phi(\Theta, \hat{\Theta}) = -\frac{1}{\alpha} \operatorname{Tr} \hat{m}^\top m + \frac{1}{2\alpha}(\operatorname{Tr} \hat{q}q + \operatorname{Tr} \hat{Q}Q) \quad (127)$$

$$+ \frac{1}{\alpha} \mathbb{E}_\varsigma Z_{kv}^*\left(\hat{m}^\top \hat{q}^{-1/2}\varsigma, \hat{m}^\top \hat{q}^{-1}\hat{m}\right) \log Z_{kv}\left(\hat{q}^{1/2}\varsigma, \hat{q} + \hat{Q}\right)$$

$$+ \mathbb{E}_L \mathbb{E}_{\xi,\varsigma} \int_{\mathbb{R}} \mathrm{d}y\, Z_{\mathrm{out}}^*\left(y, mq^{-1/2}\begin{pmatrix} \xi^\top \\ \zeta^\top \end{pmatrix}, 1 - m^\top qm\right) \log Z_{\mathrm{out}}\left(y, q^{1/2}\begin{pmatrix} \xi^\top \\ \zeta^\top \end{pmatrix}, Q - q\right)$$

$$+ \rho \mathbb{E}_L \mathbb{E}_{\xi,\varsigma} \int_{\mathbb{R}} \mathrm{d}y\, Z_{\mathrm{out}}^*\left(y, mq^{-1/2}\begin{pmatrix} \xi^\top \\ \zeta^\top \end{pmatrix}, 1 - m^\top qm\right) \log Z_{\mathrm{out}}'\left(y, q^{1/2}\begin{pmatrix} \xi^\top \\ \zeta^\top \end{pmatrix}, Q - q\right)$$

with the partition functions

$$Z_{kv}^*(B,A) = \int_{\mathbb{R}} \mathrm{d}P_k(k^*) P_v(v^*) e^{-\frac{1}{2}\binom{k^*}{v^*}^\top A \binom{k^*}{v^*} + B^\top \binom{k^*}{v^*}} \; , \tag{128}$$

$$Z_{kv}(B,A) = \int_{\mathbb{R}} \mathrm{d}\tilde{P}_k(k) \tilde{P}_v(v) e^{-\frac{1}{2}\binom{k}{v}^\top A \binom{k}{v}, + B^\top \binom{k}{v}} \tag{129}$$

$$Z_{\mathrm{out}}^*(y,\omega,V) = \int_{\mathbb{R}^L} \mathrm{d}\chi^* \mathrm{d}z^* \int \mathrm{d}P_\epsilon(\epsilon^*)\, g_\nu(\epsilon^*,\chi^*) P_{\mathrm{out}}(y|z_{\epsilon^*}^*) \prod_l^L \mathcal{N}\left(\binom{\chi_l^*}{z_l^*}; \omega_l, V\right) \tag{130}$$

$$Z_{\mathrm{out}}(y,\omega,V) = \int_{\mathbb{R}^L} \mathrm{d}\chi \mathrm{d}z \int \mathrm{d}P_\epsilon(\epsilon)\, \tilde{g}_\nu(\epsilon,\chi) \tilde{P}(y|z,\chi,\epsilon) \prod_l^L \mathcal{N}\left(\binom{\chi_l}{z_l}; \omega_l, V\right) \tag{131}$$

$$Z'_{\mathrm{out}}(\omega,V) = \int_{\mathbb{R}} \mathrm{d}\hat{y}\, e^O \int_{\mathbb{R}^L} \mathrm{d}\chi \mathrm{d}z \int \mathrm{d}P_\epsilon(\epsilon)\, \tilde{g}_\nu(\epsilon,\chi) \hat{P}(\hat{y}|z,\chi,\epsilon) \prod_l^L \mathcal{N}\left(\binom{\chi_l}{z_l}; \omega_l, V\right) \tag{132}$$

and the observables $O = sy(y-\hat{y}) + t\beta(y-\hat{y})^2$.

This expression of the free entropy can be simplified assuming that the order parameters $m,q,Q$ are diagonal. We numerically extremized $\phi$ for the full order parameters from uninformed and informed initializations and we observed that the cross-terms $m_{kv^*}, m_{vk^*}, q_{kv}, Q_{kv}$ and their conjugates go to 0. This simplification is similar to the one discussed in the main part for the population loss, though there is no direct relation with gradient descent and that we have the additional order parameter $Q$. We simplify the notations setting

$$m_k = m_{kk^*} \qquad q_k = q_{kk^*} \qquad Q_k = Q_{kk^*} \tag{133}$$
$$m_v = m_{kk^*} \qquad q_v = q_{vv^*} \qquad Q_v = Q_{vv^*} \tag{134}$$

and similarly for the conjugates. The set of order parameters becomes $\Theta = (m_k, \hat{m}_k, m_v, \hat{m}_v, q_k, \hat{q}_k, q_v, \hat{q}_v, Q_k, \hat{Q}_k, Q_v, \hat{Q}_v) \in \mathbb{R}^{12}$. The free entropy is now

$$\phi(\Theta) = -\frac{1}{\alpha}(\hat{m}_k m_k + \hat{m}_v m_v) + \frac{1}{2\alpha}(\hat{q}_k q_k + \hat{Q}_k Q_k + \hat{q}_v q_v + \hat{Q}_v Q_v) \tag{135}$$

$$+ \frac{1}{\alpha}\mathbb{E}_\varsigma Z_k^*\left(\frac{\hat{m}_k}{\sqrt{\hat{q}_k}}\varsigma, \frac{\hat{m}_k^2}{\hat{q}_k}\right) \log Z_k\left(\sqrt{\hat{q}_k}\varsigma, \hat{q}_k + \hat{Q}_k\right) + \frac{1}{\alpha}\mathbb{E}_\varsigma Z_v^*\left(\frac{\hat{m}_v}{\sqrt{\hat{q}_v}}\varsigma, \frac{\hat{m}_v^2}{\hat{q}_v}\right) \log Z_v\left(\sqrt{\hat{q}_v}\varsigma, \hat{q}_v + \hat{Q}_v\right)$$

$$+ \mathbb{E}_L \mathbb{E}_{\xi,\zeta} \int_{\mathbb{R}} \mathrm{d}y\, Z_{\mathrm{out}}^*\left(y, \frac{m_k}{\sqrt{q_k}}\xi, 1 - \frac{m_k^2}{q_k}, \frac{m_v}{\sqrt{q_v}}\zeta, 1 - \frac{m_v^2}{q_v}\right) \log Z_{\mathrm{out}}\left(y, \sqrt{q_v}\xi, Q_k - q_k, \sqrt{q_v}\zeta, Q_v - q_v\right)$$

$$+ \rho \mathbb{E}_L \mathbb{E}_{\xi,\zeta} \int_{\mathbb{R}} \mathrm{d}y\, Z_{\mathrm{out}}^*\left(y, \frac{m_k}{\sqrt{q_k}}\xi, 1 - \frac{m_k^2}{q_k}, \frac{m_v}{\sqrt{q_v}}\zeta, 1 - \frac{m_v^2}{q_v}\right) \log Z'_{\mathrm{out}}\left(\sqrt{q_v}\xi, Q_k - q_k, \sqrt{q_v}\zeta, Q_v - q_v\right)$$

with $\varsigma \sim \mathcal{N}(0,1)$ and the partition functions

$$Z_k^*(B,A) = \int_{\mathbb{R}} \mathrm{d}P_k(k^*) e^{-\frac{1}{2}A(k^*)^2 + Bk^*} \; , \qquad Z_k(B,A) = \int_{\mathbb{R}} \mathrm{d}\tilde{P}_k(k) e^{-\frac{1}{2}Ak^2 + Bk} \tag{136}$$

$$Z_v^*(B,A) = \int_{\mathbb{R}} \mathrm{d}P_v(v^*) e^{-\frac{1}{2}A(v^*)^2 + Bv^*} \; , \qquad Z_v(B,A) = \int_{\mathbb{R}} \mathrm{d}\tilde{P}_v(v) e^{-\frac{1}{2}Av^2 + Bv} \tag{137}$$

$$Z_{\mathrm{out}}^*(y,\gamma,R,\omega,V) = \int_{\mathbb{R}^L} \mathrm{d}\chi^* \mathrm{d}z^* \int \mathrm{d}P_\epsilon(\epsilon^*)\, g_\nu(\epsilon^*,\chi^*) P_{\mathrm{out}}(y|z_{\epsilon^*}^*) \prod_l^L \mathcal{N}(\chi_l^*; \gamma_l, R) \mathcal{N}(z_l^*; \omega_l, V) \tag{138}$$

$$Z_{\mathrm{out}}(y,\gamma,R,\omega,V) = \int_{\mathbb{R}^L} \mathrm{d}\chi \mathrm{d}z \int \mathrm{d}P_\epsilon(\epsilon)\, \tilde{g}_\nu(\epsilon,\chi) \tilde{P}(y|z,\chi,\epsilon) \prod_l^L \mathcal{N}(\chi_l; \gamma_l, R) \mathcal{N}(z_l; \omega_l, V) \tag{139}$$

$$Z'_{\mathrm{out}}(\gamma,R,\omega,V) = \int_{\mathbb{R}} \mathrm{d}\hat{y}\, e^O \int_{\mathbb{R}^L} \mathrm{d}\chi \mathrm{d}z \int \mathrm{d}P_\epsilon(\epsilon)\, \tilde{g}_\nu(\epsilon,\chi) \hat{P}(\hat{y}|z,\chi,\epsilon) \prod_l^L \mathcal{N}(\chi_l; \gamma_l, R) \mathcal{N}(z_l; \omega_l, V) \tag{140}$$

Once $\phi$ is extremized over the order parameters the test errors are computed as

$$\mathsf{E}_\sigma(\alpha) = \frac{1}{\beta}\mathbb{E}_L\mathbb{E}_{\xi,\varsigma}\int_\mathbb{R}\mathrm{d}y\,(Z'_\text{out})^{-1}Z_\text{out}\partial_t Z'_\text{out}|_{s=0,t=0} \tag{141}$$

$$\mathcal{E}_\text{B0} = \mathbb{E}_L\mathbb{E}_{\xi,\varsigma}\int_\mathbb{R}\mathrm{d}y\,(Z'_\text{out})^{-1}Z_\text{out}\partial_s Z'_\text{out}|_{s=0,t=0}\,. \tag{142}$$

## C.3 SPECIALIZATION TO THE BO

We consider the BO setting, where $Z_k^* = Z_k$, $Z_v^* = Z_v$ and $Z_\text{out}^* = Z_\text{out}$. A main simplification, known as Nishimori condition, is given by the fact the BO estimator is sampling the posterior distribution: we have $m_k = q_k$, $\hat{m}_k = \hat{q}_k$, $Q_k = 1$, $m_v = q_v$, $\hat{m}_v = \hat{q}_v$ and $Q_v = 1$. We set the function $\mathrm{xlogx} : x \to x\log(x)$. The resulting free entropy is

$$\phi = -\frac{1}{2\alpha}(\hat{m}_k m_k + \hat{m}_v m_v) + \frac{1}{\alpha}\mathbb{E}_\varsigma\mathrm{xlogx}Z_k\left(\sqrt{\hat{m}_k}\varsigma,\hat{m}_k\right) + \frac{1}{\alpha}\mathbb{E}_\varsigma\mathrm{xlogx}Z_v\left(\sqrt{\hat{m}_v}\varsigma,\hat{m}_v\right) \tag{143}$$

$$+ \mathbb{E}_L\mathbb{E}_{\xi,\varsigma}\int_\mathbb{R}\mathrm{d}y\,\mathrm{xlogx}Z_\text{out}\left(y,\sqrt{m}_k\xi,1-m_k,\sqrt{m_v}\zeta,1-m_v\right)$$

$$+ \rho\mathbb{E}_L\mathbb{E}_{\xi,\varsigma}\int_\mathbb{R}\mathrm{d}y\,Z_\text{out}\left(y,\sqrt{m}_k\xi,1-m_k,\sqrt{m_v}\zeta,1-m_v\right)\log Z'_\text{out}\left(\sqrt{m}_k\xi,1-m_k,\sqrt{m_v}\zeta,1-m_v\right)$$

The extremality condition of $\phi$ over the order parameters is obtained by setting its gradient to zero. We take the limit $\rho = 0$. It gives the following fixed-point equations:

$$m_k = \mathbb{E}_\varsigma Z_k^{-1}(\partial_B Z_k)^2 \tag{144}$$

$$m_v = \mathbb{E}_\varsigma Z_v^{-1}(\partial_B Z_v)^2 \tag{145}$$

$$\hat{m}_k = \alpha\mathbb{E}_L\mathbb{E}_{\xi,\varsigma}\int_\mathbb{R}\mathrm{d}y\,Z_\text{out}^{-1}(\nabla_\gamma Z_\text{out})^\top(\nabla_\gamma Z_\text{out}) \tag{146}$$

$$\hat{m}_v = \alpha\mathbb{E}_L\mathbb{E}_{\xi,\varsigma}\int_\mathbb{R}\mathrm{d}y\,Z_\text{out}^{-1}(\nabla_\omega Z_\text{out})^\top(\nabla_\omega Z_\text{out}) \tag{147}$$

We explicit the fixed-point equations. We set

$$h_\nu(\epsilon,\gamma,R) = \int_{\mathbb{R}^L}\mathrm{d}\chi\,g_\nu(\epsilon,\chi)\prod_l^L\mathcal{N}(\chi_l;\gamma_l,R) \tag{148}$$

the effective distribution over $\epsilon$. We assume that $P_\text{out}$ is the identity channel. Then

$$m_k = \frac{\hat{m}_k}{1+\hat{m}_k} \tag{149}$$

$$m_v = \frac{\hat{m}_v}{1+\hat{m}_v} \tag{150}$$

$$\hat{m}_k = \alpha\mathbb{E}_L\mathbb{E}_{\xi,\varsigma}\int_\mathbb{R}\mathrm{d}y\left[\sum_\epsilon h_\nu(\epsilon,\sqrt{m_k}\xi,1-m_k)\frac{e^{-\frac{(y-\sqrt{m_v}\zeta_\epsilon)^2}{2(1-m_v)}}}{\sqrt{2\pi(1-m_v)}}\right]^{-1} \tag{151}$$

$$\frac{1}{L}\sum_l^L\left[\sum_\epsilon\partial_{\gamma_l}h_\nu(\epsilon,\sqrt{m_k}\xi,1-m_k)\frac{e^{-\frac{(y-\sqrt{m_v}\zeta_\epsilon)^2}{2(1-m_v)}}}{\sqrt{2\pi(1-m_v)}}\right]^2$$

$$\hat{m}_v = \alpha\mathbb{E}_L\mathbb{E}_{\xi,\varsigma}\int_\mathbb{R}\mathrm{d}y\left[\sum_\epsilon h_\nu(\epsilon,\sqrt{m_k}\xi,1-m_k)\frac{e^{-\frac{(y-\sqrt{m_v}\zeta_\epsilon)^2}{2(1-m_v)}}}{\sqrt{2\pi(1-m_v)}}\right]^{-1} \tag{152}$$

$$\frac{1}{L}\sum_{l}^{L}\left[h_\nu(l,\sqrt{m_k}\xi,1-m_k)\frac{y-\sqrt{m_v}\zeta_l}{1-m_v}\frac{e^{-\frac{(y-\sqrt{m_v}\zeta_l)^2}{2(1-m_v)}}}{\sqrt{2\pi(1-m_v)}}\right]^2$$

At the fixed-point the Bayes-optimal error is given by

$$\mathcal{E}_{\text{BO}} = 1 - m_v\mathbb{E}_L\mathbb{E}_\xi\frac{1}{L}\left[\sum_\epsilon h_\nu(\epsilon,\sqrt{m_k}\xi,1-m_k)\right]^{-1}\sum_\epsilon h_\nu(\epsilon,\sqrt{m_k}\xi,1-m_k)^2 \qquad (153)$$

### C.3.1 SPECIALIZATION TO SPIKED-SLR

The spiked-SLR is defined by taking $g_\nu(\epsilon,\chi) = e^{\sqrt{\nu}\chi_\epsilon-\frac{\nu}{2}}$. We can integrate over $\chi$ and explicitly compute $h_\nu$:

$$h_\nu(\epsilon,\gamma,R) = e^{\sqrt{\nu}\gamma_\epsilon+\frac{1}{2}\nu(R-1)} . \qquad (154)$$

Thanks to the invariance by permutation of the tokens we isolate the index $l=1$. We change the variables $\xi_1 \to \xi_1 + \sqrt{\nu m_k}$ and $\zeta_1 \to \sqrt{1-m_v}\zeta_1 + \sqrt{m_v}y$ to obtain expressions that are easier to compute numerically. This gives

$$m_k = \frac{\hat{m}_k}{1+\hat{m}_k}, \qquad m_v = \frac{\hat{m}_v}{1+\hat{m}_v} \qquad (155)$$

$$\hat{m}_k = \alpha\nu\mathbb{E}_L\mathbb{E}_{\xi,\zeta}\int\frac{\mathrm{d}y}{\sqrt{2\pi}}e^{-\frac{1}{2}y^2}\frac{1}{1+\sum_{l>1}^{L}e^{-\nu m_k+\sqrt{\nu m_k}(\xi_l-\xi_1)-\frac{(y-\sqrt{m_v}\zeta_l)^2}{2(1-m_v)}+\frac{1}{2}(\sqrt{1-m_v}y-\sqrt{m_v}\zeta_1)^2}} \qquad (156)$$

$$\hat{m}_v = \frac{\alpha}{1-m_v}\mathbb{E}_L\mathbb{E}_{\xi,\zeta}\int\frac{\mathrm{d}y}{\sqrt{2\pi}}e^{-\frac{1}{2}y^2}\frac{(\sqrt{1-m_v}y-\sqrt{m_v}\zeta_1)^2}{1+\sum_{l>1}^{L}e^{-\nu m_k+\sqrt{\nu m_k}(\xi_l-\xi_1)-\frac{(y-\sqrt{m_v}\zeta_l)^2}{2(1-m_v)}+\frac{1}{2}(\sqrt{1-m_v}y-\sqrt{m_v}\zeta_1)^2}} \qquad (157)$$

$$\mathcal{E}_{\text{BO}} = 1 - m_v e^{-\frac{\nu m_k}{2}}\mathbb{E}_L\mathbb{E}_\xi\frac{e^{\sqrt{\nu m_k}\xi_1}}{1+\sum_{l>1}^{L}e^{\sqrt{\nu m_k}(\xi_l-\xi_1)}} \qquad (158)$$

### C.3.2 SPECIALIZATION TO THE MAX-SLR MODEL

The max-SLR model is defined by taking $g_\nu(\epsilon,\chi) = Le^{\nu\chi_\epsilon}/(\sum_l e^{\nu\chi_l})$. $\chi$ cannot be integrated out. We consider the limit $\nu = +\infty$ to simplify the expression of $h_\nu$:

$$h_\nu(\epsilon,\gamma,R) = L\int_\mathbb{R}\mathrm{d}\chi\,\mathcal{N}(\chi;\gamma_\epsilon,R)\prod_{l\neq\epsilon}\frac{1}{2}\left(1+\text{erf}\frac{\chi-\gamma_l}{\sqrt{2R}}\right) . \qquad (159)$$

Thanks to the invariance by permutation of the tokens we isolate the indices $l=1$ and $l=2$. We can still change variables $\zeta_\epsilon \to \sqrt{1-m_v}\zeta_\epsilon + \sqrt{m_v}y$ to obtain expressions that are easier to compute numerically. This gives

$$m_k = \frac{\hat{m}_k}{1+\hat{m}_k}, \qquad m_v = \frac{\hat{m}_v}{1+\hat{m}_v} \qquad (160)$$

$$\hat{m}_k = \alpha\mathbb{E}_L\mathbb{E}_{\xi,\zeta}\int\frac{\mathrm{d}y}{\sqrt{2\pi}}e^{-\frac{1}{2}y^2}\left[(\partial_{\gamma_1}h_\nu(1,\sqrt{m_k}\xi,1-m_k))^2e^{-\frac{1}{2}(\sqrt{1-m_v}y-\sqrt{m_v}\zeta_1)^2}\right. \qquad (161)$$

$$+ 2(L-1)\partial_{\gamma_1}h_\nu(1,\sqrt{m_k}\xi,1-m_k)\partial_{\gamma_1}h_\nu(2,\sqrt{m_k}\xi,1-m_k)e^{-\frac{(y-\sqrt{m_v}\zeta_2)^2}{2(1-m_v)}}\right]$$

$$\left[h_\nu(1,\sqrt{m_k}\xi,1-m_k)e^{-\frac{1}{2}(\sqrt{1-m_v}y-\sqrt{m_v}\zeta_1)^2} + \sum_{l>1}^{L}h_\nu(l,\sqrt{m_k}\xi,1-m_k)e^{-\frac{(y-\sqrt{m_v}\zeta_l)^2}{2(1-m_v)}}\right]^{-1}$$

$$+ \alpha\mathbb{E}_L\mathbb{E}_{\xi,\zeta}\int\frac{\mathrm{d}y}{\sqrt{2\pi}}e^{-\frac{1}{2}y^2}\left[(L-1)(\partial_{\gamma_1}h_\nu(2,\sqrt{m_k}\xi,1-m_k))^2e^{-\frac{1}{2}(\sqrt{1-m_v}y-\sqrt{m_v}\zeta_2)^2}\right.$$

$$+ (L^2 - 3L + 2)\partial_{\gamma_1} h_\nu(2, \sqrt{m_k}\xi, 1 - m_k)\partial_{\gamma_1} h_\nu(3, \sqrt{m_k}\xi, 1 - m_k)e^{-\frac{(y - \sqrt{m_v}\zeta_3)^2}{2(1 - m_v)}} \Bigg]$$

$$\left[ h_\nu(2, \sqrt{m_k}\xi, 1 - m_k)e^{-\frac{1}{2}(\sqrt{1 - m_v}y - \sqrt{m_v}\zeta_2)^2} + \sum_{l \neq 2}^L h_\nu(l, \sqrt{m_k}\xi, 1 - m_k)e^{-\frac{(y - \sqrt{m_v}\zeta_l)^2}{2(1 - m_v)}} \right]^{-1}$$

$$\hat{m}_v = \alpha \mathbb{E}_L \mathbb{E}_{\xi,\zeta} \int \frac{dy}{\sqrt{2\pi}} e^{-\frac{1}{2}y^2} \frac{(\sqrt{1 - m_v}y - \sqrt{m_v}\zeta_1)^2}{1 - m_v} h_\nu(1, \sqrt{m_k}\xi, 1 - m_k)e^{-\frac{1}{2}(\sqrt{1 - m_v}y - \sqrt{m_v}\zeta_1)^2}$$

$$\left[ h_\nu(1, \sqrt{m_k}\xi, 1 - m_k)e^{-\frac{1}{2}(\sqrt{1 - m_v}y - \sqrt{m_v}\zeta_1)^2} + \sum_{l > 1}^L h_\nu(l, \sqrt{m_k}\xi, 1 - m_k)e^{-\frac{(y - \sqrt{m_v}\zeta_l)^2}{2(1 - m_v)}} \right]^{-1}$$

$$\tag{162}$$

$$\mathcal{E}_{\text{BO}} = 1 - m_v \mathbb{E}_L \mathbb{E}_\xi \frac{1}{L} h_\nu(1, \sqrt{m_k}\xi, 1 - m_k)^2 \tag{163}$$

## C.4 SPECIALIZATION TO THE ATTENTION

We consider the attention case, where the distributions $\tilde{P}_k$, $\tilde{P}_v$ and $\tilde{P}(y|X, \epsilon, k, v)$ do not match the distributions of the model. In this case it is more convenient to work with the variables

$$V_k = Q_k - q_k , \quad \hat{V}_k = \hat{Q}_k + \hat{q}_k , \quad V_v = Q_v - q_v , \quad \hat{V}_v = \hat{Q}_v + \hat{q}_v . \tag{164}$$

We perform the changes $\varsigma_k \to \varsigma_k + k^* \hat{m}_k / \sqrt{\hat{q}_k}$, $\varsigma_v \to \varsigma_v + v^* \hat{m}_v / \sqrt{\hat{q}_v}$, $\xi_l \to \xi_l \sqrt{q_k - m_k^2} / \sqrt{q_k} + \chi_l^* m_k / \sqrt{q_k}$ and $\zeta_l \to \zeta_l \sqrt{q_v - m_v^2} / \sqrt{q_v} + z_l^* m_v / \sqrt{q_v}$. We deal with the limit $\beta \to \infty$ by rescaling the parameters according to $\hat{m}_k \to \beta \hat{m}_k$, $\hat{m}_v \to \beta \hat{m}_v$, $\hat{q}_k \to \beta^2 \hat{q}_k$, $\hat{q}_v \to \beta^2 \hat{q}_v$, $\hat{V}_k \to \beta \hat{V}_k$, $\hat{V}_v \to \beta \hat{V}_v$ and $V_k \to \beta^{-1} V_k$, $V_v \to \beta^{-1} V_v$.

We introduce the effective joint distribution of the data, for $y \in \mathbb{R}$, $\epsilon^* \in \{1, \dots, L\}$, $\chi^* \in \mathbb{R}^L$ and $z^* \in \mathbb{R}^L$:

$$P^*(y, \epsilon^*, \chi^*, z^*) = \frac{1}{L} g_\nu(\epsilon^*, \chi^*) P_{\text{out}}(y|z_{\epsilon^*}^*) \prod_l^L \mathcal{N}(\chi_l^*; 0, 1)\mathcal{N}(z_l^*; 0, 1) . \tag{165}$$

The free entropy is then

$$\frac{1}{\beta}\phi(\Theta) = -\frac{1}{\alpha}(\hat{m}_k m_k + \hat{m}_v m_v) + \frac{1}{2\alpha}\left( \frac{1}{\beta} V_k \hat{V}_k + q_k \hat{V}_k - V_k \hat{q}_k + \frac{1}{\beta} V_v \hat{V}_v + q_v \hat{V}_v - V_v \hat{q}_v \right)$$

$$+ \frac{1}{\alpha\beta}\mathbb{E}_\varsigma \int_\mathbb{R} dP_k(k^*) \log \int_\mathbb{R} dk \, e^{\beta\psi_k(k)} + \frac{1}{\alpha\beta}\mathbb{E}_\varsigma \int_\mathbb{R} dP_v(v^*) \log \int_\mathbb{R} dv \, e^{\beta\psi_v(v)} \tag{166}$$

$$+ \frac{1}{\beta}\mathbb{E}_L \mathbb{E}_{\xi,\zeta} \int dP^*(y, \epsilon^*, \chi^*, z^*) \log \int_{\mathbb{R}^\mathbb{L}} d\chi dz \left( e^{\beta\psi_{\text{out}}(\chi, z; 1)} + \rho e^{\beta\psi_{\text{out}}(\chi, z; 0) + t\beta(y - \hat{y}_\sigma(z, \chi))^2} \right)$$

with the potentials

$$\psi_k(k) = -r_k \gamma(k) - \frac{1}{2}\hat{V}_k k^2 + (\hat{m}_k k^* + \sqrt{\hat{q}_k}\varsigma)k \tag{167}$$

$$\psi_v(v) = -r_v \gamma(v) - \frac{1}{2}\hat{V}_v v^2 + (\hat{m}_v v^* + \sqrt{\hat{q}_v}\varsigma)v \tag{168}$$

$$\psi_{\text{out}}(\chi, z; \bar{t}) = -\bar{t}\ell(y, \hat{y}_\sigma(z, \chi)) + \sum_l^L \log \mathcal{N}(\chi_l; \gamma_l, V_k) + \sum_l^L \log \mathcal{N}(z_l; \omega_l, V_v) \tag{169}$$

$$\gamma = m_k \chi^* + \sqrt{q_k - m_k^2}\xi , \quad \omega = m_v z^* + \sqrt{q_v - m_v^2}\zeta \tag{170}$$

$\bar{t} \in \{0, 1\}$ controls whether the loss or the observable are active or not. We introduce the extremizers of these potentials:

$$k' = \arg\max_k \psi_k(k) , \quad v' = \arg\max_v \psi_v(v) \tag{171}$$

$$\chi', z' = \arg\max_{\chi, z} \psi_{\text{out}}(\chi, z; \bar{t} = 1), \qquad \chi'', z'' = \arg\max_{\chi, z} \psi_{\text{out}}(\chi, z; \bar{t} = 0) \tag{172}$$

We introduce the covariances under these potentials around there maxima, with $\nabla^2$ the Hessian.

$$\text{Cov}(k) = -\left(\nabla^2 \psi_k(k')\right)^{-1}, \qquad\qquad \text{Cov}(v) = -\left(\nabla^2 \psi_v(v')\right)^{-1} \tag{173}$$

$$\text{Cov}(\chi_l) = -\left(\left(\nabla^2 \psi_{\text{out}}(\chi', z'; \bar{t} = 1)\right)^{-1}\right)_{\chi_l}, \quad \text{Cov}(z_l) = -\left(\left(\nabla^2 \psi_{\text{out}}(\chi', z'; \bar{t} = 1)\right)^{-1}\right)_{z_l} \tag{174}$$

The extremality condition of $\phi$ over the order parameters is obtained by setting its gradient to zero. We take the limit $\rho = 0$. It gives the following fixed-point equations:

$$m_k = \mathbb{E}_\varsigma \int_{\mathbb{R}} dP_k(k^*) \, k^* k' \qquad\qquad m_v = \mathbb{E}_\varsigma \int_{\mathbb{R}} dP_v(v^*) \, v^* v' \tag{175}$$

$$q_k = \mathbb{E}_\varsigma \int_{\mathbb{R}} dP_k(k^*) \, (k')^2 \qquad\qquad q_v = \mathbb{E}_\varsigma \int_{\mathbb{R}} dP_v(v^*) \, (v')^2 \tag{176}$$

$$V_k = \mathbb{E}_\varsigma \int_{\mathbb{R}} dP_k \, \text{Cov}(k) \qquad\qquad V_v = \mathbb{E}_\varsigma \int_{\mathbb{R}} dP_v \, \text{Cov}(v) \tag{177}$$

and

$$\hat{m}_k = \alpha \mathbb{E}_L \mathbb{E}_{\xi, \varsigma} \int dP^*(y, \epsilon^*, \chi^*, z^*) \frac{1}{V_k} \sum_l^L \left(\chi_l^* \chi_l' - \frac{m_k}{V_k} \text{Cov}(\chi_l)\right) \tag{178}$$

$$\hat{m}_v = \alpha \mathbb{E}_L \mathbb{E}_{\xi, \varsigma} \int dP^*(y, \epsilon^*, \chi^*, z^*) \frac{1}{V_v} \left(z_{\epsilon^*}^* z_{\epsilon^*}' - \frac{m_v}{V_v} \text{Cov}(z_{\epsilon^*})\right) \tag{179}$$

$$\hat{q}_k = \alpha \mathbb{E}_L \mathbb{E}_{\xi, \varsigma} \int dP^*(y, \epsilon^*, \chi^*, z^*) \frac{1}{V_k^2} \sum_l^L \left(\chi_l' - m_k \chi_l^* - \sqrt{q_k - m_k^2} \xi_l\right)^2 \tag{180}$$

$$\hat{q}_v = \alpha \mathbb{E}_L \mathbb{E}_{\xi, \varsigma} \int dP^*(y, \epsilon^*, \chi^*, z^*) \frac{1}{V_v^2} \sum_l^L \left(z_l' - m_v z_l^* - \sqrt{q_v - m_v^2} \zeta_l\right)^2 \tag{181}$$

$$\hat{V}_k = \mathbb{E}_L \frac{\alpha L}{V_k} - \alpha \mathbb{E}_L \mathbb{E}_{\xi, \varsigma} \int dP^*(y, \epsilon^*, \chi^*, z^*) \frac{1}{V_k^2} \sum_l^L \text{Cov}(\chi_l) \tag{182}$$

$$\hat{V}_v = \mathbb{E}_L \frac{\alpha L}{V_v} - \alpha \mathbb{E}_L \mathbb{E}_{\xi, \varsigma} \int dP^*(y, \epsilon^*, \chi^*, z^*) \frac{1}{V_v^2} \sum_l^L \text{Cov}(z_l) \tag{183}$$

At the fixed-point the test error is given by

$$\mathsf{E}_\sigma(\alpha) = \mathbb{E}_L \mathbb{E}_{\xi, \varsigma} \int dP^*(y, \epsilon^*, \chi^*, z^*) \, (y - \hat{y}_\sigma(\chi'', z''))^2. \tag{184}$$

### C.4.1  SPECIALIZATIONS TO SPIKED- AND MAX-SLR MODELS

We consider an attention trained with $l_2$ regularization $\gamma(x) = \frac{1}{2}x^2$. The first fixed-point equations can be explicited

$$m_k = \frac{\hat{m}_k}{r_k + \hat{V}_k} \qquad\qquad m_v = \frac{\hat{m}_v}{r_v + \hat{V}_v} \tag{185}$$

$$q_k = \frac{\hat{m}_k^2 + \hat{q}_k}{(r_k + \hat{V}_k)^2} \qquad\qquad q_v = \frac{\hat{m}_v^2 + \hat{q}_v}{(r_v + \hat{V}_v)^2} \tag{186}$$

$$V_k = \frac{1}{r_k + \hat{V}_k} \qquad\qquad V_v = \frac{1}{r_v + \hat{V}_v} \tag{187}$$

We can simplify the rest of the equations by using the permutation invariance w.r.t. the tokens to fix $\epsilon^* = 1$. For the two models we have that, for $l \neq \epsilon^*$, $z_l^*$ is Gaussian under $P^*(y, \epsilon^*, \chi^*, z^*)$ and

$m_v z_l^* + \sqrt{q_v - m_v^2}\zeta_l \sim \mathcal{N}(0, q_v)$, so we can integrate out $z_l^*$. We assume that $P_{\text{out}}$ is the identity channel so $z_{\epsilon^*}^* = y$. In the potential $\psi_{\text{out}}$ we have $\omega_l = m_v y \delta_{l,1} + \sqrt{q_v - m_v^2 \delta_{l,1}}\zeta_l$ for all $l$. The joint distribution over data reduces to

$$P^*(y, \epsilon^*, \chi^*, z^*) = P^*(y, \chi^*) = \mathcal{N}(y; 0, 1) g_\nu(1, \chi^*) \prod_l^L \mathcal{N}(\chi_l^*; 0, 1). \tag{188}$$

Taking $y \sim \mathcal{N}(0, 1)$ and $\chi^* \sim \mathcal{N}(0, I_L)$, the fixed-point equations and the error are

$$\hat{m}_k = \alpha \mathbb{E}_L \mathbb{E}_{\xi, \zeta, y, \chi^*} g_\nu(1, \chi^*) \frac{1}{V_k} \sum_l^L \left( \chi_l^* \chi_l' - \frac{m_k}{V_k} \operatorname{Cov}(\chi_l) \right) \tag{189}$$

$$\hat{m}_v = \alpha \mathbb{E}_L \mathbb{E}_{\xi, \zeta, y, \chi^*} g_\nu(1, \chi^*) \frac{1}{V_v} \left( y z_1' - \frac{m_v}{V_v} \operatorname{Cov}(z_1) \right) \tag{190}$$

$$\hat{q}_k = \alpha \mathbb{E}_L \mathbb{E}_{\xi, \zeta, y, \chi^*} g_\nu(1, \chi^*) \frac{1}{V_k^2} \sum_l^L \left( \chi_l' - m_k \chi_l^* - \sqrt{q_k - m_k^2}\xi_l \right)^2 \tag{191}$$

$$\hat{q}_v = \alpha \mathbb{E}_L \mathbb{E}_{\xi, \zeta, y, \chi^*} g_\nu(1, \chi^*) \frac{1}{V_v^2} \left( \left( z_l' - m_v y - \sqrt{q_v - m_v^2}\zeta_l \right)^2 + \sum_{l>1}^L (z_l' - \sqrt{q_v}\zeta_l)^2 \right) \tag{192}$$

$$\hat{V}_k = \mathbb{E}_L \frac{\alpha L}{V_k} - \alpha \mathbb{E}_L \mathbb{E}_{\xi, \zeta, y, \chi^*} g_\nu(1, \chi^*) \frac{1}{V_k^2} \sum_l^L \operatorname{Cov}(\chi_l) \tag{193}$$

$$\hat{V}_v = \mathbb{E}_L \frac{\alpha L}{V_v} - \alpha \mathbb{E}_L \mathbb{E}_{\xi, \zeta, y, \chi^*} g_\nu(1, \chi^*) \frac{1}{V_v^2} \sum_l^L \operatorname{Cov}(z_l) \tag{194}$$

$$\mathsf{E}_\sigma(\alpha) = \mathbb{E}_L \mathbb{E}_{\xi, \zeta, y, \chi^*} g_\nu(1, \chi^*)(y - \hat{y}_\sigma(\chi'', z''))^2 \tag{195}$$

For the spiked-SLR model we additionally have that $\chi_l^*$ is Gaussian under $P^*(y, \chi^*)$; so $m_k \chi_l^* + \sqrt{q_k - m_k^2}\xi_l \sim \mathcal{N}(\sqrt{\nu}m_k \delta_{l,1}, q_k)$ and $\chi^*$ can be integrated out. In the potential $\psi_{\text{out}}$ we have $\gamma_l = \sqrt{\nu}m_k \delta_{l,1} + \sqrt{q_v}\xi_l$ for all $l$. We have the simplifications

$$\hat{m}_k = \alpha \mathbb{E}_L \mathbb{E}_{\xi, \zeta, y} \frac{1}{V_k} \left( \sqrt{\nu}\chi_1' - \nu m_k \right) \tag{196}$$

$$\hat{q}_k = \alpha \mathbb{E}_L \mathbb{E}_{\xi, \zeta, y} \frac{1}{V_k^2} \left( (\chi_1' - \sqrt{\nu}m_k - \sqrt{q_k}\xi_1)^2 + \sum_{l>1}^L (\chi_l' - \sqrt{q_k}\xi_l)^2 \right) \tag{197}$$

# D  DETAILS ON THE NUMERICS

## D.1  TRAINING OF THE ATTENTION

The attention eq. (4) is trained using LBFGS on the full batch until convergence. The parameters $v$ and $k$ are initialized randomly according to a standard Gaussian $\mathcal{N}(0, I_D)$.

In Fig. 3 the regularizations $r_k$ and $r_v$ are tuned by grid search to minimize the test error, over $\{0.3, 1, 3, 10\}^2$ for the linear attention and $\{0.03, 0.1, 0.3, 1\}^2$ for the softmax. These values were determined according to Fig. 7.

For most of the parameters of the data model we explored the training converges to the global minimum of the training loss eq. (15), in the sense that the test risk of trained attention is equal to the test risk predicted by our asymptotic characterization.

In a few cases, in particular at low signal and low regularization, the training does not converge towards the global minimum of the loss, which results in a discrepancy between the simulated performances and the predicted ones. In these cases we train the attention starting from an informed initialization $v = v^*$ and $k = k^*$. Note that this initialization does not necessary correspond to a minimum of the loss. The performances of the trained attention then better matches our predictions, as depicted in Fig. 7 for the softmax on the max-SLR. We checked that the achieved training loss is well smaller than the one starting from a random initialization.

## D.2  COMPUTATION OF THE ASYMPTOTIC CHARACTERIZATION

### D.2.1  BO

We compute the BO performance stated in result B.1 by iterating the fixed point equations given by the condition $\nabla \phi_{\mathrm{BO}}(m_k, m_v) = 0$. These equations are detailed in appendices C.3.1 and C.3.2 for the two models. The uninformed initialization corresponds to $m_k, m_v = 0.1, 0.1$ while the informed initialization corresponds to $m_k, m_v = 1 - 10^{-2}, 1 - 10^{-2}$. The expectation over the Gaussian random variables is computed over $10^6$ and $10^5$ Monte-Carlo samples respectively for the spiked-SLR and the max-SLR models.

### D.2.2  ATTENTION

We compute the performance of the attention stated in result 5.1 by iterating the given equations. We use Steffensen's method to speed up the convergence. For the spiked-SLR a simplification of the equations is given in Appendix C.4.1. The expectation over the Gaussian random variables is computed over $10^5$ Monte-Carlo samples. For each sample at each iteration the extremizers of $\psi_{\mathrm{out}}$ are computed using a quasi-Newton optimization scheme. The optimization is started at the extremizers computed at the previous step.

Notice that $\psi_{\mathrm{out}}$ is not convex and may admit several maxima. In practice, for most of the values of $\gamma, \omega, V_k$ and $V_v$, $\psi_{\mathrm{out}}$ admits a unique maximum. When it is not the case (in particular at low signal and low regularization) one should compare several different initializations of the optimization algorithm to find the global maximum. We tried on a few cases; it appeared that the final predictions do not change by a quantity greater than the fluctuations due to the randomness.

The minimum over the set $\mathcal{S}$ of fixed points is computed by running the iterations of result 5.1 from several initializations. We considered informed initialization $(m_k, m_v, q_k, q_v, V_k, V_v) = (1, 1, 0, 0, \varepsilon, \varepsilon)$ for small $\varepsilon$, partially informed initializations $(m_k, m_v, q_k, q_v, V_k, V_v) = (1, 1, 1, 1, 1, 1)$ and uninformed initializations $(m_k, m_v, q_k, q_v, V_k, V_v) = (0, 0, 1, 1, 1, 1)$. We performed a few different runs, adding small randomness to the initial condition. For all the values of the parameters, the obtained fixed point did not depend on the choice of the initialization, up to some small numerical fluctuations. These fluctuations are larger at small regularizations and low signals, in which case we select the run which reaches the highest free entropy eq. (166), i.e. the lowest train loss, among those that converged before a certain amount of iterations.

# E    ADDITIONAL FIGURES

We provide an additional figure for the result 5.1 about the test risk of the attention at finite $\alpha$. Fig. 7 gives the test risk $\mathsf{E}_\sigma$ for the same configurations as in Fig. 3 for several additional regularizations $r_k$ and $r_v$. It justifies the ranges of the grid search for the linear and the softmax attention in Fig. 3, in the sense that $\mathsf{E}_\sigma$ seems to reach its minimal value for regularizations close to the ones considered in Fig. 7. Moreover it shows the excellent agreement between our theory result 5.1 and the simulations. We observe a discrepancy for the softmax at small regularization for the spiked-SLR at $\nu = 1$ or the max-SLR at $\nu = +\infty$. For the spiked-SLR it may be caused by the replica-symmetry assumption being false or the numerical errors in the resolution of the fixed point equations. For the max-SLR at $r_k = r_v = 0.03$ we observe that initializing the simulations at the informed point $k = k^*, v = v^*$ leads to an agreement with our theory. This shows that in this case the local optimization cannot recover the global minimum of the loss from random initialization.

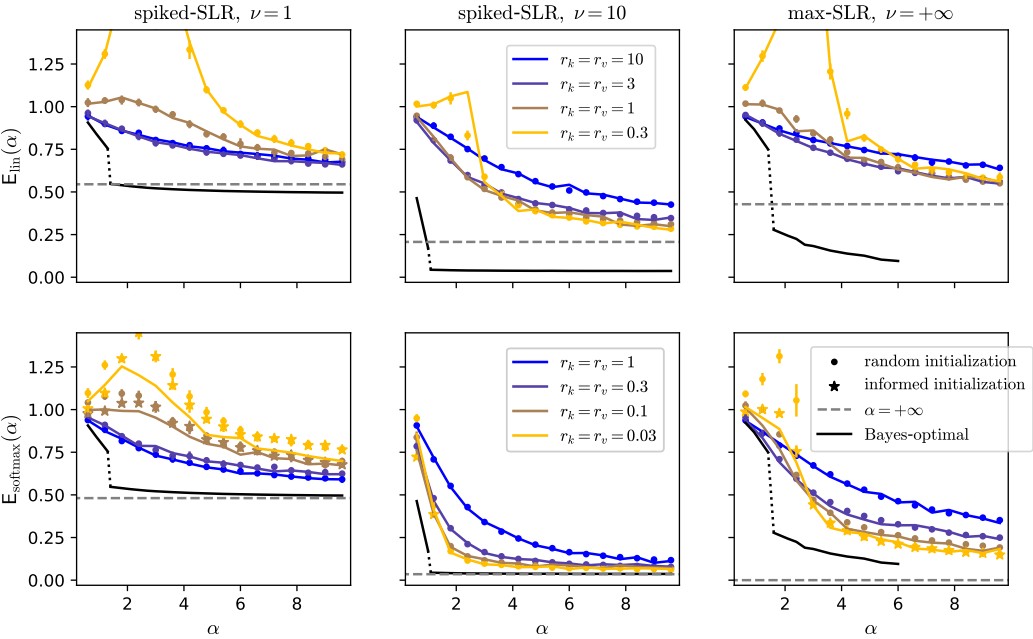

Figure 7: Minimal test risk $\mathsf{E}_\sigma(\alpha)$ of the attention, linear (top) and softmax (bottom), across different tasks and signal strengths $\nu$, for $L = 3$. Solid lines indicate $\mathsf{E}_\sigma(\alpha)$ (Result 5.1), while markers represent the test risk of an ERM approximated via a local optimization method with $\sqrt{ND} = 10^4$ and averaged over ten instances. Random initialization means that the attention is initialized at random $k$ and $v$ while informed initialization means it is initialized at $k = k^*, v = v^*$. Dashed lines correspond to the value of $\mathsf{E}_\sigma$ in the infinite-$\alpha$ limit (see closed-formed formulas in Proposition 4.1 for softmax and Appendix A.2.7 for linear). The Bayes-optimal risk $\mathcal{E}_{\mathrm{BO}}(\alpha)$ is shown in black (see Section B for a discussion on its discontinuity). Appendix D includes more experimental details.

## ACKNOWLEDGMENTS

We used large language models for helping the writing and discussing some references and related work.

