# OpenReview forum: "Statistical Advantage of Softmax Attention: Insights from Single-Location Regression"
_ICLR.cc/2026/Conference — ICLR 2026 Poster_

### Official Review · Reviewer_J4sf · 2025-11-01

**Soundness:** 3
**Presentation:** 3
**Contribution:** 2
**Rating:** 4
**Confidence:** 2

**Summary:**

The paper studies why softmax attention often outperforms linear attention by analyzing a stylized but mathematically tractable setting called Single-Location Regression (SLR). In this setup, only one randomly chosen token in a sequence carries useful information for predicting the target. The authors derive population-risk expressions showing that softmax attention achieves the Bayes risk (i.e., the minimum possible expected loss), while linear attention is inherently suboptimal. They further extend the analysis to finite-sample training using replica methods and confirm that gradient-based optimization empirically follows the predicted risk curves. The work provides a clear theoretical explanation for the statistical advantage of softmax normalization.

**Strengths:**

- Provides a principled and analytically clean setup (SLR) that isolates the retrieval aspect of attention mechanisms.

- Derives explicit population-risk expressions demonstrating when and why softmax outperforms linear attention.

- Extends the analysis to the finite-sample regime using replica theory, connecting asymptotic analysis with realistic training outcomes.

- Numerical simulations show strong alignment with theoretical predictions, suggesting the model captures key phenomena of attention mechanisms.

- Offers valuable theoretical insight into the long-standing question of softmax’s statistical advantage, which is of interest to both theoretical and applied communities.

**Weaknesses:**

1. Limited task scope. The analysis focuses solely on the single-location regression (SLR) task, which captures a narrow form of retrieval behavior and does not generalize to multi-token dependencies or compositional reasoning.

2. Restricted model comparison. The study only contrasts softmax and linear attention, omitting other relevant variants such as kernelized softmax approximations and state-space models (SSMs), which prevents a broader understanding of where softmax’s advantage truly lies.

3. Reliance on idealized assumptions. The theoretical results depend on Gaussian i.i.d. token embeddings and replica-symmetry assumptions; robustness to more realistic distributions and correlated features is not explored.

4. Absence of realistic experiments. All evaluations are conducted on synthetic data, with no validation on real-world or language-based datasets, leaving uncertain whether the observed statistical gap translates into practical performance gains.

**Questions:**

Please refer to the weakness section

---

> ### Author Response · Authors · 2025-11-21
>
> We thank the reviewer for the comments and suggestions.
>
> W1: please see the common answer.
>
> W2: our study does not only contrast softmax and linear attentions. In Section 4.4 and in Figure 1 we propose a comparison with other activation functions than linear and softmax, including a kernelized softmax, which sheds light onto softmax’s advantage.
>
> W3: for the reliance on Gaussian i.i.d. token embeddings, please see the common answer. For the reliance on the replica-symmetry assumption: this is not an idealization, but instead a technical assumption needed to derive Result 5.1; it is justified by the excellent agreement of the predictions of Result 5.1 with the simulations shown on Figure 3.
>
> W4: please see the common answer.

---

### Official Review · Reviewer_hz39 · 2025-11-01

**Soundness:** 3
**Presentation:** 3
**Contribution:** 2
**Rating:** 6
**Confidence:** 2

**Summary:**

This paper studies the theoretical foundations of softmax attention by introducing a principled single-location regression (SLR) model. The authors derive analytical and asymptotic results using statistical physics tools (notably order parameters and replica analysis) to show that softmax attention achieves Bayes-optimal performance in high-dimensional limits, while linear attention and other alternatives (e.g., kernelized or element-wise nonlinearities) fundamentally fall short. The paper further provides a finite-sample characterization of test risk, confirming the statistical and computational advantages of softmax. Overall, it offers a clean theoretical framework that bridges information retrieval toy models and high-dimensional generalization analysis.

**Strengths:**

Clear and original theoretical contribution:
The paper introduces a new analytical framework — the Single-Location Regression (SLR) model — to study the statistical behavior of attention mechanisms. This formulation unifies previous “needle-in-a-haystack”-type setups under a mathematically tractable regime, enabling a principled comparison between softmax and linear attention.

Methodological depth:
The work combines sequence multi-index models with replica-based high-dimensional analysis, bringing together tools from statistical physics and modern learning theory. This cross-disciplinary approach is technically sophisticated and extends recent progress in the theoretical understanding of attention networks.

**Weaknesses:**

Idealized task setting:
The SLR model assumes the label depends on a single token’s linear transformation, which, while analytically convenient, is far from the multi-head, multi-layer structure of real Transformers. Hence, the practical relevance is limited.

Limited empirical grounding:
The validation is restricted to synthetic setups, without experiments on realistic retrieval or sequence tasks. It remains unclear whether the predicted statistical advantage of softmax can be observed in real neural models.

**Questions:**

Can the proposed Single-Location Regression framework be extended to multi-location or multi-head attention, where multiple tokens jointly determine the output? Would the Bayes-optimality of softmax still hold in those cases?

The analysis assumes Gaussian and independent token embeddings. How sensitive are the results to these assumptions? Would correlated or structured embeddings affect the theoretical conclusions?


Have the authors tested whether the predicted statistical gap between softmax and linear attention appears in small-scale Transformer experiments or synthetic retrieval tasks (e.g., Needle-in-a-Haystack)?

---

> ### Author Response · Authors · 2025-11-21
>
> We thank the reviewer for the comments and suggestions.
>
> W1: please see the common answer.
>
> W2: please see the common answer.
>
> Q1: the Single-Location Regression framework can be extended to multi-location or multi-head attention, in the line of study of the sequence multi-index models (Cui 2025). This is out of the scope of the present paper, and such an analysis first requires to understand the single-location case, which as we show already presents a rich phenomenology.
>
> Q2: please see the common answer.
>
> Q3: please see the common answer.

---

### Official Review · Reviewer_Uy5X · 2025-11-02

**Soundness:** 2
**Presentation:** 2
**Contribution:** 3
**Rating:** 4
**Confidence:** 3

**Summary:**

This paper theoretically explains why softmax attention outperforms linear attention by analyzing a single-location regression task  (Marion et al., 2025), where output depends on one hidden token, proving softmax achieves Bayes-optimal risk while linear attention fundamentally fails.

Building on sequence multi-index models (Cui et al. 2024, Troiani et al. 2025), the authors extend these techniques to variable-length sequences and characterize both population and finite-sample risks via replica methods, showing that exponential nonlinearity and normalization are both critical.

Experiments validate theoretical predictions with agreement between predicted and observed test error curves.

**Strengths:**

The authors provide theoretical results establishing provable performance gaps between softmax and linear attention ( by analyzing a single-location regression task ) at the population level (Propositions 4.1-4.4, Corollaries 4.2-4.3).


The paper successfully extends sequence multi-index model techniques to handle the challenging softmax nonlinearity through careful renormalization and order parameter analysis.

The authors provide strong empirical validation. Code was shared but I didn't check.  The experiments provide quantitative agreement between theoretical predictions and gradient-based optimization experiments across multiple settings (different signal strengths ν, sequence lengths L, regularizations).

**Weaknesses:**

- No optimization theory/landscape analysis is provided, and this is indeed a limitation as the paper proves softmax's empirical risk minimizer has superior statistical properties but provides no characterization of the optimization landscape or guarantees connecting estimation to optimization. The empirical risk (15) is non-convex with multiple local minima (Appendix A.2.6), yet the paper offers no analysis of landscape geometry, convergence guarantees for gradient descent, or the estimation-optimization gap.

- The main finite-sample characterization relies on the replica symmetry assumption from statistical physics, which lacks rigorous justification. This should be stated more prominently in the theorem statement itself, not buried in discussion.


- Expectation notation: $E[\cdot]$ vs. $\mathbb{E}[\cdot]$ inconsistently used
- Risk terminology: switches between "test error," "test risk," "Bayes risk," "Bayes error," and "Bayes-optimal risk".

- "Symmetric activation function" in Corollary 4.1 needs precise definition.


- Equation (3) notation confusion: The notation $\chi^* = (1/\sqrt{D}) x \otimes k^*$ uses $\otimes$ which typically denotes tensor product, but context suggests matrix multiplication. Should be clarified or corrected.

- Proposition 4.2, condition (6): States "for all $L > 0$" but $L$ is sequence length (positive integer), should be "for all $L \ge 1$" or "for all $L \in \mathbb{N}$".

- Asymptotic notation in eq. (12): The notation $E_{\text{lin}} = (L/(L-1))(1/\nu) + o_{\nu\to\infty}(1)$ places subscript on little-o, which is non-standard. Should be $E_{\text{lin}} = (L/(L-1))(1/\nu) + o(1)$ as $\nu \to \infty$.

- High-dimensional limit not precisely stated: The relationship between $N$, $D$, and $\alpha = N/D$ in the limit is ambiguous. Is it $D \to \infty$ then $N \to \infty$? Or jointly? The order and dependence should be explicit early in the paper.

- Manifold assumption justification: The manifold $\mathcal{M}$ is introduced abruptly on page 5. While Section A.2.2 shows it's invariant under gradient descent, more intuitive geometric/statistical interpretation would help readers understand why this restriction is natural.


-  Equations (18)–(25)  present complex fixed-point equations with no intuitive explanation. What do these order parameters represent physically? Why these specific equations?

- Figure 1 misleading markers: Caption states "markers on the lines are for readability only." If markers don't represent actual data points, they should be removed as they're potentially misleading.

- Figure 2 incomplete experimental details: Caption mentions regularizations "tuned by grid search" but doesn't specify the grid values in the experiments section.


- Some figure legends are too small to read comfortably (especially Figure 3).

**Questions:**

Please see Weaknesses.

---

> ### Author Response · Authors · 2025-11-21
>
> We thank the reviewer for the comments and suggestions.
>
> Optimization theory/landscape analysis: we actually do provide analysis of the landscape and the optimization. Specifically, we prove that the manifold $\mathcal{M}$ that we introduce is invariant by gradient flow. We also provide a guarantee of convergence for the linear attention towards its global minimizer on $\mathcal{M}$. To make this point clearer we have added an additional corollary in the main text. Finally, both for the linear and softmax attention, on fig. 3 we verify numerically that gradient-based optimization algorithms reach the global minimum we describe. Taken together, these facts support our analysis focusing on the statistical properties of the global minimum. A full landscape analysis is left for future work, as stated in Section 4.2.
>
> Replica symmetry assumption: we stated it more clearly in the result. As to the assumption itself, it is justified by the excellent agreement of the predictions of Result 5.1 with the simulations shown on Figure 3.
>
> Expectation notation: we use $\mathbb{E}[\cdot]$ for expectation, could the reviewer more precisely point to where we use $E[\cdot]$ for expectation? We use $E_\sigma$ to denote the minimized risk as now summarized in a terminology table in the updated manuscript.
>
> Risk terminology: we added a table to summarize the terminology and different notations we use for the risks. We improved the clarity by limiting the number of used synonyms.
>
> For the symmetric activation function, the precise definition is already stated in Corollary 4.1.
>
> We do not use the tensor notation $\otimes$ anywhere in the paper. Could the referee point to a line which motivated this question?
>
> High-dimensional limit: we further precise at the beginning of Section 5 how the limit between $N$ and $D$ should be taken. We do not discuss this before in the paper, since the finite sample size $N$ is only introduced in Section 5, while the previous sections focus on the population risk.
>
> Manifold assumption justification: the intuition behind introducing the manifold is that, in the SLR, the signals $k^\*$ and $v^\*$ associated to the keys and the values are drawn independent; and to obtain good performance, the keys and the values should not mix and should focus on $k^\*$ or on $v^\*$ separately. We added this explanation after the introduction of the manifold.
>
> Intuitive explanation of the self-consistent equations: to give some intuition we provide an explanation of what the order parameters mean; please consider the revised version of the manuscript after result 5.1.
>
> For Figure 1, the markers are for accessibility, in case the colors cannot be distinguished, or when the curves are superimposed.
>
> For Figure 2 the parameters of the grid search are detailed in appendix D, as now specified in the caption of the figure.
>
> Figure 3: we updated the figure, enlarging the labels and the legend.

---

### Official Review · Reviewer_tjje · 2025-11-03

**Soundness:** 3
**Presentation:** 3
**Contribution:** 3
**Rating:** 6
**Confidence:** 2

**Summary:**

In this paper, the authors do a study to show why softmax-based attention mechanism outperforms alternative attention variants (such as linear attention). To do so, they use a simplified “single-location regression” task in a theoretical framework. They show that in the infinite-data population limit, a softmax attention model achieves the Bayes-optimal prediction error, which cannot be reached by linear attention models. Furthermore, they also show, that even in the finite case, where no model can reach the Bayes risk, softmax attention still outperforms linear attention.

**Strengths:**

I think the paper has the following strengths:

1) Principled Task Formalization - the paper introduces a clear mathematical framework for studying attention mechanisms by formalizing a single-location regression model, in which the output depends on a single informative token in the input sequence. In this way, this setup generalized earlier theoretical studies by allowing random sequence lengths and mirrorying “needle-in-a-haystack” retrieval tasks.

2) Rigorous Theoretical Analysis and Key Insight - The authors develop a tractable high-dimensional analysis of attention layers using techniques from sequence mulit-index models that the softmax nonlinearity can be handles with a small set of parameters. What is quite exciting is that they show that compared to linear attention, only the softmax attention can achieve the Bayes-optimal error, while the linear attention is suboptimal.

3) Comprehensive Evaluation of Performance - the study also examines the finite-sample regime (the real-world case) adding practical value to the theoretical analysis in (2). They show that the softmax model still holds an advantage compared to linear attention in the finite-sample setting.

**Weaknesses:**

I think the paper can be further improved in these parts:

1) Limited Scope of the Task and Model - the analysis in limited to the simplified single-location retrieval scenario. While this abstraction (which corresponds to using a fixed query token like [CLS]) is mathematically convenient and appropriate for the task, it does not cover cases where multiple tokens or more complex query mechanisms are involved. Real-world Transformers often deal with interactions among many relevant tokens and use learned queries, so I am not sure how the softmax advantage might play out in these scenarios.

2) Gap to Real-World Validation -  The paper’s findings are supported primarily by theory and synthetic experiments. All experiments are on simplified tasks defined by the authors (e.g. variants of the single-location regression or synthetic retrieval scenarios), which raises the question of how well the insights transfer to practical large-scale NLP tasks. As a result, the work stops short of empirically confirming that softmax’s advantage in more complex or realistic conditions.

**Questions:**

Slightly off the topic, but recently, there has been an interest in softmax-1 activation [A, B, C], which is a version of softmax designed to prevent attending in the first (few tokens). Considering the similarities between it and the standard softmax, I wonder if the findings would hold there.

[A] Miller, Attention is off by one, 2023

[B]  Bondarenko et al., Quantizable Transformers: Removing Outliers by Helping Attention Heads Do Nothing, NeurIPS, 2023

[C] Kaul et al., From Attention to Activation: Unravelling the Enigmas of Large Language Models, ICLR 2025

---

> ### Author Response · Authors · 2025-11-21
>
> We thank the reviewer for the comments and suggestions.
>
> About the weaknesses, please see the common answer.
>
> About the softmax-1 activation: we thank the reviewer for the pointer. The softmax-1 is designed to allow attention heads not to attend to any token. In our case, it is not the most appropriate activation because there is always a token the attention should attend to. For this reason, it will not reach the Bayes risk. But the softmax-1 definitely seems an interesting direction for future research.

---

### Author Response · Authors · 2025-11-21
**Common answer**

We thank the reviewers for their comments and for highlighting the quality, originality, and anticipated importance of our article. We uploaded a revised version of the manuscript implementing the suggestions, and below we answer the common remarks or questions the reviewers raised.

**Idealized task setting, limited task scope:** we agree that understanding compositional reasoning, multi-token dependencies or more complex query mechanisms are major goals. Yet, the choice of our model, the SLR task, is not a weakness, for the following reasons:
* On a theoretical point of view, the analysis we carried is challenging enough so that it has never been carried before. According to reviewer *hz39*, it is a clear and original theoretical contribution. When aiming to understand multi-token models, it is a clear prerequisite to have a detailed understanding of a single-token dependency model first. As reviewers *J4sf* and *tjje* highlight, the SLR provides a principled and analytically clean setup. Our analysis can be extended to a multi-token setting using the same tools and the same framework, with a considerable aditional work and some new technical elements; as such, the present paper is the first necessary step toward analyzing more complicated tasks and architectures.
* The SLR task gives a clear explanation of the better performances of the softmax, over linear or kernalized attention, which is a crucial topic for transformers and LLMs. As such, it is of interest to both theoretical and applied communities, as highlighted by reviewer *J4sf*. We believe that presenting a minimal setting where an advantage of a widely used ingredient -- such as softmax -- can be  clearly demonstrated is scientifically valuable. Our opinion is that considering other more complex settings in a single paper would dilute the message and the presented intuition. But we agree that those are all interesting questions and we certainly hope that our paper will motivate ample follow-up work in these directions.
* As we show, even a single-token dependency presents a rich phenomenology with multiple minima of the risk, that was not previously described.

**Extension to more realistic distributions and correlated features:** our work can definitively be adapted to embeddings that have a richer structure. One can assume a covariance with a well-defined asymptotic spectral density, or a hidden random manifold on which the features are embedded, and adapt the equations of https://journals.aps.org/prx/abstract/10.1103/PhysRevX.10.041044 for one-pass GD to the attention on the SLR. We expect that the phenomenology would be different and dependent on the structure of the correlations. For instance, if all the tokens have a larger covariance in a direction aligned with the signals $k^\*$ and $v^\*$, we can expect a mismatch where the values $v$ align with $k^\*$ and the keys $k$ with $v^\*$.

**Validation on realistic experiments:** we appreciate the reviewers' emphasis on empirical validation in realistic settings. In the case at hand, a substantial body of prior work has already documented the empirical advantages of softmax attention over several alternatives, including kernelized-attention methods and state-space models. Because these results are well established, we focused our contribution on the theoretical aspects rather than attempting to replicate findings that are already consistent across the literature. That said, we agree that explicitly summarizing this evidence strengthens the motivation for our study. In the revised manuscript, we provide a more comprehensive overview of these empirical results, including a detailed discussion of the experiments  by Shen et al. (2024). In brief, their findings indicate that while optimally scaled SSMs and kernelized-attention models achieve competitive performance on linguistic proficiency benchmarks, they consistently underperform softmax attention on retrieval-oriented tasks. Again, we would like to stress that the fact that SLR-type tasks have been widely benchmarked in the literature attests its relevance to practice.

---

### Meta-Review · Area_Chair_kbgA · 2026-01-06

**Summary:**

The authors study a toy model of needle-in-a-haystack (NIAH) which they call single-location regression (SLR), where the goal is to essentially perform regression but based on a single token only, and the location of the token has to be learned. In principle, it is a simplification of the usual NIAH task, but it does capture the difficulty of the location aspect for the task.

The reviewers found strengths in:
- a unified and mathematical tractable framework for NIAH retrieval tasks
- a strong asymptotic analysis of attention for softmax and other forms of non-linearities
- strong numerical experiments verifying the theory

On the other hand the reviewers found weaknesses in:
- limited to the idealized setting with narrow scope
- lacks validation on realistic settings

Overall, I find myself more sympathetic to the authors. I believe the authors are attempting to find the equivalent of a single/multi-index model for retrieval tasks, so that one can study the complexity of learning with attention mechanisms. From this point of view, the results on the proportional limit are quite nice. In particular, the authors demonstrated that there is a gap between information theoretic limit and the computational complexity using first order algorithms.

My only minor critiques are that:
- the toy model does not involve a query term, only a key term
- the calculations in the proportional regime are using replica methods
- there is not description of algorithmic learning

Personally, none of these are critical issues, as they are simply expanding scope of the research. I think the current set of results are sufficient for a submission at ICLR.

To conclude, I think the reviews are somewhat harsher than the submission deserve, and I am very comfortable with recommending accept.

**Reviewer Concerns:**

Limited to the idealized setting with narrow scope
- partially addressed in the sense the authors defend the modelling choice, unclear the reviewer response

Lacks validation on realistic settings
- not addressed in the sense that no experiments were added

**Reviewer Scores:**

Reviewer tjje: 6
Reviewer Uy5X: 4
Reviewer hz39: 6
Reviewer J4sf: 4

---

### Decision · Program_Chairs · 2026-01-26

Accept (Poster)